# Computational modelling identifies key determinants of subregion-specific dopamine dynamics in the striatum

Aske Ejdrup[1†], Jakob Kisbye Dreyer[2], Matthew D Lycas[1], Søren H Jørgensen[1], Trevor W Robbins[3,4], Jeffrey Dalley[3,4,5], Freja Herborg[1*†], Ulrik Gether[1*†]

[1]Department of Neuroscience, Faculty of Health and Medical Sciences, University of Copenhagen, Copenhagen, Denmark; [2]Department of Bioinformatics, H Lundbeck A/S, Valby, Denmark; [3]Behavioural and Clinical Neuroscience Institute, University of Cambridge, Cambridge, United Kingdom; [4]Department of Psychology, University of Cambridge, Cambridge, United Kingdom; [5]Department of Psychiatry, University of Cambridge, Cambridge, United Kingdom

*For correspondence:
frejahh@sund.ku.dk (FH);
gether@sund.ku.dk (UG)

[†]These authors contributed equally to this work.

## eLife Assessment

The conclusions of this work are based on **valuable** simulations of a detailed model of striatal dopamine dynamics. Establishing that lower dopamine uptake rate can lead to a "tonic" level of dopamine in the ventral but not dorsal striatum, and that dopamine concentration changes at short delays can be tracked by D1 but not D2 receptor activation, is invaluable and will be of interest to the community, particularly those studying dopamine. The model simulations provide **convincing** evidence for differences between dorsal and ventral striatum dopamine concentrations, while evidence for differential tracking of dopamine changes by D1 vs D2 receptors is **solid**.

**Abstract** Striatal dopamine (DA) release regulates reward-related learning and motivation and is believed to consist of a short-lived *phasic* and continuous *tonic* component. Here, we build a large-scale three-dimensional model of extracellular DA dynamics in dorsal (DS) and ventral striatum (VS). The model predicts rapid dynamics in DS with little to no basal DA and slower dynamics in the VS enabling build-up of *tonic* DA levels. These regional differences do not reflect release-related phenomena but rather differential dopamine transporter (DAT) activity. Interestingly, our simulations posit DAT nanoclustering as a possible regulator of this activity. Receptor binding simulations show that D1 receptor occupancy follows extracellular DA concentration with milliseconds delay, while D2 receptors do not respond to brief pauses in firing but rather integrate DA signal over seconds. Summarised, our model distills recent experimental observations into a computational framework that challenges prevailing paradigms of striatal DA signalling.

## Introduction

Striatal dopamine (DA) release is essential for regulating reward-related learning, incentive motivation, and motor function (*Berke, 2018*; *Klaus et al., 2019*). DA exerts these roles over a broad range of time scales, yet DA primarily operates as a volume transmitter that targets metabotropic receptors located within a micrometre range from the sites of release (*Agnati et al., 1995*; *Borroto-Escuela et al., 2018*; *Cragg and Rice, 2004*; *Gonon et al., 2000*; *Sulzer et al., 2016*). The temporal and spatial dynamics of DA release in the striatum, however, remain a highly contested topic. Classically, DA release has been divided into *tonic* release, driven by pacemaker-like spontaneous firing, and

*phasic* release from coordinated bursts of firing across neurons (*Niv et al., 2007*; *Schultz, 2007*; *Sulzer et al., 2016*). However, this sharp distinction in release modes, as well as the existence of a basal DA level, has recently been challenged (*Berke, 2018*; *Ejdrup et al., 2023*; *Jørgensen et al., 2023*; *Liu et al., 2021*; *Sippy and Tritsch, 2023*).

The picture is further complicated by major regional differences across striatal subdomains. These include differences in $Ca^{2+}$-channel and nicotinic acetylcholine receptor (nAChR) expression profiles on DA terminals, as well as differential regulation and expression of the DA transporter (DAT; *Brown et al., 2011*; *Cardozo and Bean, 1995*; *Kearney et al., 2023*; *Richards and Zahniser, 2009*; *Threlfell et al., 2010*). In addition, we and others have found remarkable differences in extracellular DA release dynamics between the dorsal (DS) and ventral striatum (VS; *Jørgensen et al., 2023*; *Mohebi et al., 2024*; *Salinas et al., 2023*). Fibre photometry recordings in the DS in mice using the DA sensor dLight1.3b during self-paced exploratory activity showed a rapidly fluctuating signal, whereas we observed up to minutes-long DA dynamics in VS that correlated with behavioural output (*Jørgensen et al., 2023*). Concurrent measurements of extracellular DA by microdialysis and fibre photometry have furthermore corroborated the lack of tonic levels of DA in DS while supporting its presence in VS (*Ejdrup et al., 2023*; *Jørgensen et al., 2023*). Despite these reported differences in striatal DA dynamics, electrophysiological recordings suggest that DA neurons from the primary innervators of DS, substantia nigra par compacta (SNc) and VS, ventral tegmental area (VTA), have remarkably similar firing patterns at rest (*Dodson et al., 2016*). We therefore set out to better understand the fundamental principles governing extracellular DA dynamics by constructing a new computational model of the striatal DA system.

Extracellular DA dynamics have been modelled before; either one-dimensionally or with a primary focus on single release events or post-synaptic receptor binding (*Beyene et al., 2017*; *Dreyer et al., 2010*; *Dreyer and Hounsgaard, 2013*; *Dreyer et al., 2016*; *Venton et al., 2003*; *Wiencke et al., 2020*). Here, we present a three-dimensional model of tens of thousands of release sites, focused on larger-scale signalling and based on experimentally observed biological parameters. The model faithfully replicates experimentally observed results as well as the difference in DA dynamics between DS and VS. Importantly, it offers compelling evidence that these differences do not primarily reflect different release phenomena but rather arise from differential expression and possibly nanoscale localisation of the DAT.

## Results
### Construction of a model of DA dynamics in the striatum

We constructed a novel model of DA release using experimentally determined parameters from DS, including release, uptake, and cytoarchitecture (*Doucet et al., 1986*; *Dreyer et al., 2010*; *Dreyer and Hounsgaard, 2013*; *Liu et al., 2021*; *Olson et al., 1972*; *Sulzer et al., 2016*). DA release sites on axons projecting from the midbrain were randomly simulated as uniformly distributed discrete points in a three-dimensional space (*Figure 1A*). The release events themselves were simulated as point source events (*Cragg and Rice, 2004*) driven by action potentials (AP). We then modelled DA release for each voxel in the simulation containing a release site as a function of three key parameters: firing rate, release probability, and quantal size:

$$\text{release}_{n,t} = \text{Poisson}\left(f_{rate}\,dt\right)_n P\left(R_\%\right)_t Q\,dt \qquad (1)$$

where $\text{Poisson}\left(f_{rate}\,dt\right)_n$ is a Poisson distribution of action potentials (AP) for the given neuron (n) with the firing rate $f_{rate}(l)$, $P\left(R_\%\right)_t$ is the probability of release at the individual terminal (t) for each AP, while $Q$ is the number of DA molecules released per event (dopaminergic quantal size) and $dt$ the time step. Changing $f_{rate}$ can be used to model both pacemaker firing, typically reported at 2–10 Hz, and burst firing, which can exceed 20 Hz (*Sulzer et al., 2016*).

DA reuptake in the striatum is almost exclusively mediated by the DAT (*Jones et al., 1998*), which is widely distributed along DA axons and varicosities (*Block et al., 2015*; *Eriksen et al., 2010*; *Eriksen et al., 2009*). As reuptake follows concentration-dependent Michaelis-Menten kinetics (*Nicholson, 1995*), we simulated uptake as follows:

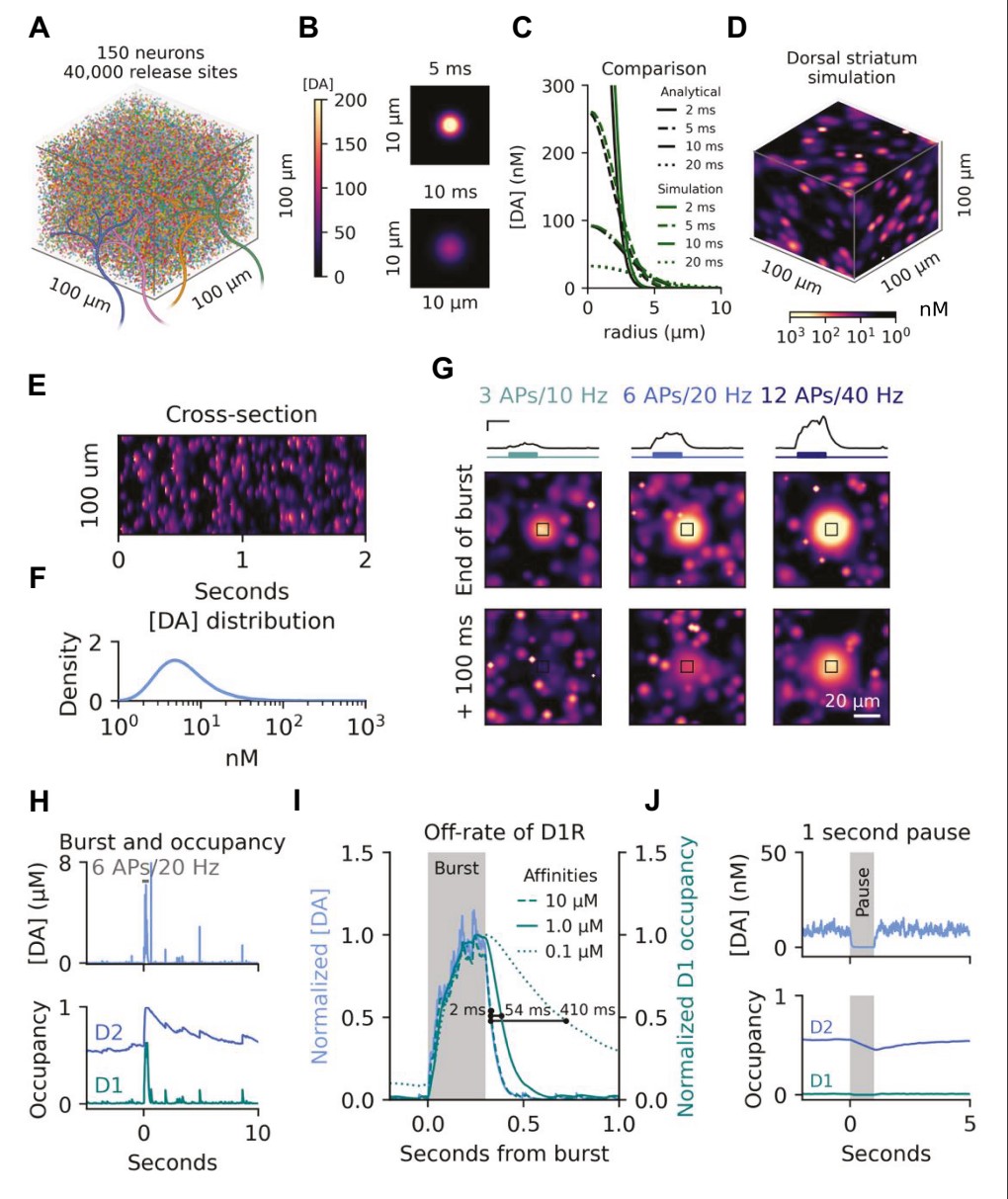

**Figure 1.** Large-scale 3D model of the dorsal striatum. (**A**) Self-enveloped simulation space of 100 μm³ with approximately 40,000 release sites from 150 neurons. Colours of individual release sites are not matched to neurons. (**B**) Simulation of a single release event after 5 ms and 10 ms. Colour-coded by DA concentration. (**C**) Comparison of analytical solution and simulation of diffusion after a single release event at three different time points. (**D**) Representative snapshot of steady state DA dynamics at 4 Hz tonic firing with parameters mirroring the dorsal striatum. (**E**) Cross-section of temporal dynamics for a midway section through the simulation space shown in (**d**). (**F**) Histogram of DA concentrations ([DA]) across the entire space in (**d**). (**G**) DA release during three burst activity scenarios for all release sites in a 10 x 10 × 10 μm cube (black boxes) and spill-over into the surrounding space. Burst simulated as an increase in firing rate on top of continued tonic firing of the surrounding space. Traces on top are average DA concentrations for the marked cubes, with bursts schematised by coloured lines below. The first image row is at the end of the burst, and the second row is 100 ms after. Scale bars for traces are 200 ms and 500 nM. Scale bar for the images is 20 μm. (**H**) Top: representative [DA] trace for a voxel with a release site during pacemaker and burst activity. Bottom: Occupancy of D1Rs and D2Rs for the same site. (**I**) Zoom on a DA burst as in (**h**), with [DA] in blue and D1R occupancy in teal with line style indicating different affinities. The shaded area indicates the period of bursting with 6 APs at 20 Hz. (**J**) Effect of complete pause in firing for 1 s on both average [DA] and D1R and D2R occupation.

The online version of this article includes the following video, source code, and figure supplement(s) for figure 1:

*Figure 1 continued on next page*

*Figure 1 continued*

**Source code 1.** Source code used to generate data in A, D, E, and F.

**Source code 2.** Source code used to generate data in B and C.

**Source code 3.** Source code used to generate data in G.

**Source code 4.** Source code used to generate data in J.

**Source code 5.** Source code used to generate data in H.

**Source code 6.** Source code used to generate data in I.

**Figure supplement 1.** Average concentration and receptor kinetics.

**Figure supplement 1—source code 1.** Source code used to generate data in *Figure 1—figure supplement 1*.

**Figure supplement 2.** Simulation size and granularity.

**Figure supplement 2—source code 1.** Source code used to generate data in *Figure 1—figure supplement 2*.

**Figure 1—video 1.** Representative video of steady-state dynamics at 4 Hz tonic firing in a 100 x 100 × 100 µm volume with parameters mirroring those experimentally observed in dorsal (left) and ventral (right) striatum.
https://elifesciences.org/articles/105214/figures#fig1video1

**Video 1—source code 1.** Source code used to generate data in Figure 1—video 1 and Figure 1—video 2.

**Figure 1—video 2.** Representative video of a cross-section of a burst firing of 6 action potentials (AP) at 20 Hz the centre of the plane (circle) for the dorsal (upper row) and ventral (bottom row) striatum during steady state dynamics at 4 Hz tonic firing.
https://elifesciences.org/articles/105214/figures#fig1video2

$$\text{uptake} = \frac{V_{max}\left[DA\right]}{K_m + \left[DA\right]}\,dt \tag{2}$$

where $\left[DA\right]$ is the concentration of DA for each voxel in the model, $V_{max}$ is the maximal uptake capacity in the region and $K_m$ is the concentration of DA at which half of $V_{max}$ is reached.

The spatial distribution of released DA is a complex interplay between release, uptake, and diffusion. Diffusion in an open 3D space can be simulated for each voxel with a Laplacian operator:

$$\text{diffusion} = \frac{\partial DA_{x,y,z,t}}{\partial t} = D_a dt \left( \frac{\partial^2 DA_{x,y,z,t}}{\partial x^2} + \frac{\partial^2 DA_{x,y,z,t}}{\partial y^2} + \frac{\partial^2 DA_{x,y,z,t}}{\partial z^2} \right) \tag{3}$$

where $D_a$ is a corrected diffusion coefficient and *dt* is the timestep. As the extracellular space of the striatum is tortuous, we modified the conventional diffusion coefficient $D$ to an apparent diffusion coefficient ($D_a$) to correct for the tortuosity ($\lambda$) of the striatum (*Cragg and Rice, 2004*; *Nicholson, 1985*):

$$D_a = \frac{D}{\lambda^2} \tag{4}$$

As the cerebellum exhibits a tortuosity similar to that recorded in the striatum, we assumed a uniform tortuosity throughout the striatum (*Nicholson and Phillips, 1981*). Combining *Equations 1–4*, we model DA changes in each voxel with a single conceptual equation:

$$\frac{dDA}{dt} = \text{release} - \text{uptake} + \text{diffusion} \tag{5}$$

We first compared our 3D model of DA dynamics to the analytical solution of a single release event (*Cragg and Rice, 2004*; *Gonon et al., 2000*). To do this, we simulated a quantal event of 3000 DA molecules and calculated DA concentrations across space at three separate time points (*Figure 1B and C*). The analytic solution and our model predicted almost identical results. Slight differences were introduced as the analytical solution assumes linear uptake from DAT while our model incorporates non-linear Michaelis-Menten kinetics. These differences, however, were almost negligible. The main difference between the two models lies in scalability across both space and time. Summarised, the model enables a dynamic incorporation of the surrounding DA concentration, release events, and uptake and can be scaled to cover DA dynamics of a large 3D space, whose size and granularity is

**Table 1.** List of variables used in the simulation of the dorsal striatum.

| Variable | Abbreviation | Value | Reference |
|---|---|---|---|
| Firing rate | | 4 Hz | *Paladini et al., 2003* |
| Release probability | | 6% | *Dreyer et al., 2010* |
| DA molecules per vesicle | | 3000 | *Klaus et al., 2019* |
| Diffusion coefficient | | 763 $\mu m^2 s^{-1}$ | *Nicholson, 1995* |
| Tortuosity | | 1.54 | *Rice and Nicholson, 1991* |
| Vmax | | 6.0 $\mu m\ s^{-1}$ | See *Appendix 2—table 2* |
| Km | | 210 nM | *Hovde et al., 2019* |
| Active terminal density | - | 0.04 $\mu m^{-3}$ | *Liu et al., 2021* |
| Extracellular volume fraction | | 0.21 | *Rice and Nicholson, 1991* |
| Number of neurons in simulation space | - | 150 | *Matsuda et al., 2009* |

only limited by computing power (see Code Availability Section for the Python code with numerical implementations of the equations listed and scripts to run the simulations and plot the main figures).

We also tested the validity of our model by examining the response to electrical stimulation. Importantly, our model faithfully mirrored DA release seen with fast-scan cyclic voltammetry (FSCV) recordings upon direct stimulation of striatal slices when we corrected for kinetics of the typical FSCV recording setup (*Figure 1—figure supplement 1A, B* and Appendix 1 - supplementary text) (*Atcherley et al., 2015*; *Brimblecombe et al., 2019*; *Stuber et al., 2010*; *Xie et al., 2020*).

## Simulating large-scale DA dynamics of the dorsal striatum

To better understand the extracellular DA dynamics that arise from the balance between dopaminergic pacemaker activity and uptake, we simulated DA dynamics in the DS generated by pacemaker activity (4 Hz) of 150 neurons in the midbrain in a 100 x 100 × 100 μm space (*Figure 1D*, see parameters in *Table 1*). Our simulations yielded a pattern of partially segregated DA hot spots with large fractions of the simulated space devoid of DA, suggesting that release events in DS only elevate DA in the immediate surroundings, with DAT-dependent clearance preventing a larger spread in space (*Figure 1D* and *Figure 1—video 1*). This was also illustrated by a cross-section in time (*Figure 1E*). In line with our recent in vivo microdialysis experiments, the average DA concentration in the simulations during pacemaker activity was approximately 10 nM (*Ejdrup et al., 2023*). Further, when we calculated the average concentration of a larger area across time, which fibre photometry conceivably does, the results resembled a tonic DA concentration (*Figure 1—figure supplement 1C*). However, our model predicted a spatial distribution that is highly heterogenous and devoid of pervasive resting or tonic DA levels (*Figure 1D–F*).

To ensure our simulations were performed within a sufficiently large space to yield consistent results, we tested different sizes of the simulated area and found a diameter of 50 μm to faithfully mimic the results of larger simulations (*Figure 1—figure supplement 2A, B*). Additionally, we tested our simulations at different granularity (0.1, 0.5, 1, and 2 μm). The finer the spatial grain, the higher the detail close to a release event; however, at a spatial granularity of 1 μm, [DA] deviated by <2% across most percentiles and only by >1 nM above the 99.5$^{th}$ percentile (*Figure 1—figure supplement 2C-F*), leading us to use this voxel size for our simulations.

## Burst firing and receptor occupancy

DA neurons are known to fire short bursts of APs, which is a phenomenon strongly linked to reward-prediction error and learning (*Schultz, 2007*). These bursts can also be induced locally in the striatum by nicotinic receptor activation (*Liu et al., 2022*; *Matityahu et al., 2023*). To gain insights into extracellular DA dynamics following a locally induced burst, we simulated three different firing scenarios for a group of terminals within a 10 x 10 × 10 μm field encompassing roughly 40 release sites from the randomly simulated 150 neurons: 3 pulses at 10 Hz, 6 pulses at 20 Hz, and 12 pulses at 40 Hz (*Figure 1G* – burst properties matched to be the same duration).

The middle scenario most closely resembles the physiological burst behaviour reported in the literature, whereas the high-activity burst is above what is typically seen. Unsurprisingly, peak DA concentration was reached at the end of the bursts (*Figure 1G*). The 3 APs/10 Hz bursting scenario generated no significant spill-over of DA outside the region of activity, whereas the 6 APs/20 Hz and 12 APs/40 Hz bursting scenarios markedly overwhelmed uptake (*Figure 1G*). The relationship between firing rate and the sphere of influence by DA became further evident when plotting maximal concentration of the surrounding space (*Figure 1—figure supplement 1D*) and the volume of space with a DA concentration above 100 nM (*Figure 1—figure supplement 1E*). We found that the 3 APs/10 Hz stimulation produced DA responses that largely resembled that of a single pulse. In both cases, DA was mostly cleared after 100ms and the volume exposed to greater than 100 nM was similar (*Figure 1G*, *Figure 1—figure supplement 1E*). In contrast, the high bursting activities caused a frequency-dependent spill-over, where the areas exposed to a DA concentration above 100 nM were 10 and 30 times larger than the terminal origin for 6 APs/20 Hz and 12 APs/40 Hz, respectively. Even after 100ms, a considerable amount of DA remained in the 12 APs/40 Hz scenario (*Figure 1G*).

To understand how these DA dynamics could affect the postsynaptic response, we modelled receptor binding. D1 receptors (-Rs) were assumed to have a half maximal effective concentration ($EC_{50}$) of 1000 nM, and we extrapolated the reverse rate constant ($k_{off}$) to 19.5 s$^{-1}$ based on a linear fit of the recently characterised DA-receptor-based sensors (*Figure 1—figure supplement 1E*; *Labouesse and Patriarchi, 2021*). We set the $EC_{50}$ of D2Rs to 7 nM and $k_{off}$ to 0.2 s$^{-1}$ based on the DA sensor kinetic fit (*Figure 1—figure supplement 1F*), which matches a recent binding study (0.197 s$^{-1}$ for binding study vs. 0.204 s$^{-1}$ based on linear fit), indicating the receptor-based sensor fit can be extrapolated to the endogenous receptors (*Ågren et al., 2021*). To determine how these receptors would respond to our predicted DA dynamics, we simulated pacemaker activity at 4 Hz with an added burst of 6 APs/20 Hz. *Figure 1H* shows a representative trace of DA concentration and occupancy of the D1R and D2R for a voxel with a release site. During pacemaker activity, D1R showed an occupancy close to 0, whereas D2R occupancy was approximately 0.55 (*Figure 1H*). Both D1R and D2R occupancies were due to a high diffusion rate mostly invariant to individual release events caused by pacemaker activity. However, upon coordinated burst firing, the occupancy rapidly increased (*Figure 1H* and *Figure 1—video 2*) as diffusion no longer equilibrates the extracellular concentrations on a timescale faster than the receptors. D1R receptor occupancy closely tracked extracellular DA with a delay of only ~50 ms for the typically reported affinity of 1 μM (*Figure 1I*). By contrast, it took at least 5 s before the burst-induced increase in D2R occupancy had declined to baseline levels (*Figure 1H* and *Figure 1—video 2*). This made the D2R incapable of temporally separating closely linked bursts of activity and rather summarised the output, whereas the D1R occupancy reset between each individual burst (*Figure 1—figure supplement 1G*). Perhaps more surprisingly, the D2R occupancy only fell from approximately from 0.55 to 0.45 when simulating a full second pause in firing due to the slow off kinetics (*Figure 1J*). Indeed, this finding was robust across an order of magnitude of D2R affinity (2 nm - 20 nM), although the sensitivity to a one-second pause was larger at an affinity of 20 nM (*Figure 1—figure supplement 1H*).

These simulations suggest that the dopaminergic architecture of the DS limits DA overflow during physiologically relevant bursting activity. Further, DA receptors had a temporally mostly uniform response to DA release caused by pacemaker activity, with D1R occupancy responding rapidly to both onset and offset extracellular DA concentrations following bursts, while D2R showed seconds-long delays in offset.

## Ventral striatum maintains pervasive DA tone

Mounting evidence points to considerable differences in DA dynamics across striatal subregions (*Jørgensen et al., 2023*; *Mohebi et al., 2024*), which might reflect differences in the cytoarchitectural and/or molecular dopaminergic makeup. In line with this, most studies report lower dopaminergic density in the VS than in DS regardless of methodological modality with a median value of ~90% in VS relative to DS (*Appendix 2—table 1*). Further, DAT-mediated uptake capacity is reported to be lower in VS with a median capacity at ~30% of DS (*Appendix 2—Tables 1 and 2*). Consistently, we observed a clear dorsoventral gradient for DAT expression when analysing immunostainings in striatal mouse brain slices from a previous publication (*Sørensen et al., 2021*; *Figure 2—figure supplement 1A, B*).

By contrast, the VMAT2 staining only decreased slightly from VS to DS (*Figure 2—figure supplement 1A–C*).

We simulated DA release during pacemaker activity in both DS and VS. DS values were set as previously described (25 $\mu m^3$ per terminal, uptake capacity of 6.0 $\mu M\ s^{-1}$), but for VS we reduced the terminal density to 90% (27.8 $\mu m^3$ per terminal) and DAT uptake capacity to 33% (2.0 $\mu M\ s^{-1}$) (*Appendix 2—Tables 1 and 2*). The remaining parameters were kept identical. With these two differences, our simulations revealed markedly different spatiotemporal DA distributions during pacemaker activity. While DS formed segregated domains with low DA concentrations in the inter-domain space (*Figure 2A* and *Figure 1—video 1*), DA diffused further throughout the simulated space in VS, before being cleared by DAT. This gave rise to what may be considered a tonic DA level with hotspots of higher DA concentrations, although the concentration distribution is continuous (*Figure 2A–C* and *Figure 1—video 1*).

We compared our model of the two regions with existing experimental data. In an earlier study by May and Wightman, 120 stimulus pulses were delivered in the medial forebrain bundle (MFB) at either 10, 30, or 60 Hz and DA responses were recorded by FSCV in both caudate-putamen (CPu) and nucleus accumbens (Nac; *May and Wightman, 1989*). To mirror this, we simulated 120 action potentials at similar frequencies (10, 30, and 60 Hz) at 6% release probability and ran the result through convolution, as in *Figure 1—figure supplement 1A and B*, to generate an FSCV read-out (*Figure 2E*). Since May and Wightman reported no significant difference in DA released per electrically delivered pulse ([DA]$_p$) between VS and DS, we applied equal quantal size and R$_%$ for DS and VS in our simulations, while uptake capacity in VS was kept to a third of DS and terminal density was set to 90% as specified above. Importantly, our simulated FSCV data closely resembled the earlier findings, with VS reaching considerably higher DA levels for all three stimulation frequencies (*Figure 2E* – see *May and Wightman, 1989*). This regional difference presumably arises from differences in DAT capacity between DS and VS, as the lower terminal density in VS would have the opposite effect (see below) and the remaining parameters were held identical.

To compare with our results for DS, we tested how VS responded during simulated burst activity. Using firing patterns identical to the DS simulations (*Figure 1G*), we found a larger spill-over of DA into the surrounding areas in VS (*Figure 2F*, *Figure 2—figure supplement 1D, E*). Significant amounts of extracellular DA also remained 100 ms after the physiologically relevant 6 APs/20 Hz firing stimulus. At the receptor level, D1R occupancy in VS showed a similar response to that in DS during the burst (*Figures 1I and 2G* and *Figure 1—video 2*). By contrast, D2R occupancy during pacemaker activity was higher in VS than DS (~0.8 versus ~0.55 in DS), in accordance with the higher prevailing basal DA concentration. Additionally, the larger DA overflow in VS after a burst caused a higher relative increase in receptor occupancy further away from the area actively bursting than compared to DS (*Figure 2H*). A pause in firing had the same effect on D2R as in DS (*Figure 2—figure supplement 1F*).

## Changes to uptake capacity greatly affect [DA] in the ventral striatum

The values used to model the striatum (*Table 1*) in the previous simulations were chosen to best mimic the physiological system found in vivo. However, to test the robustness of the results, we performed simulations across wide ranges of the variable key parameters on which the model is based. First, we varied the number of varicosities actively releasing DA by setting the varicosity density to one site per 9 $\mu m^3$ (*Doucet et al., 1986*) and simulating 4 Hz pacemaker activity with the release-capable fraction ranging from 5% to 100% (reported values range from 20% to virtually all) (*Ducrot et al., 2021*; *Liu et al., 2021*; *Liu et al., 2018*; *Pereira et al., 2016*; *Figure 3A*). As the fraction of active sites increased, DA concentrations increased at both the median level (50th percentile), which we consider a measure of tonic or baseline DA levels, and at peak levels (99.5th percentile) in both DS and VS (*Figure 3B*, see *Figure 3—figure supplement 1A* for schematic of tonic and peak DA). We then used the 99.5th/50th percentile ratio as a measure of the focality of the DA distribution (i.e. hotspot DA relative to baseline DA). This was intended as a measure of heterogeneity, that is, the higher focality, the greater competence for spatially heterogenous signalling, as has been reported in *Hamid et al., 2021*; *Howe and Dombeck, 2016*. Quantifying this across the percentage of active terminals showed that the focality of the DA distribution dropped as the active fraction increased in both regions (*Figure 3C*). However, the percentage of active sites in VS needed to drop to 5% to reach a relative distribution resembling

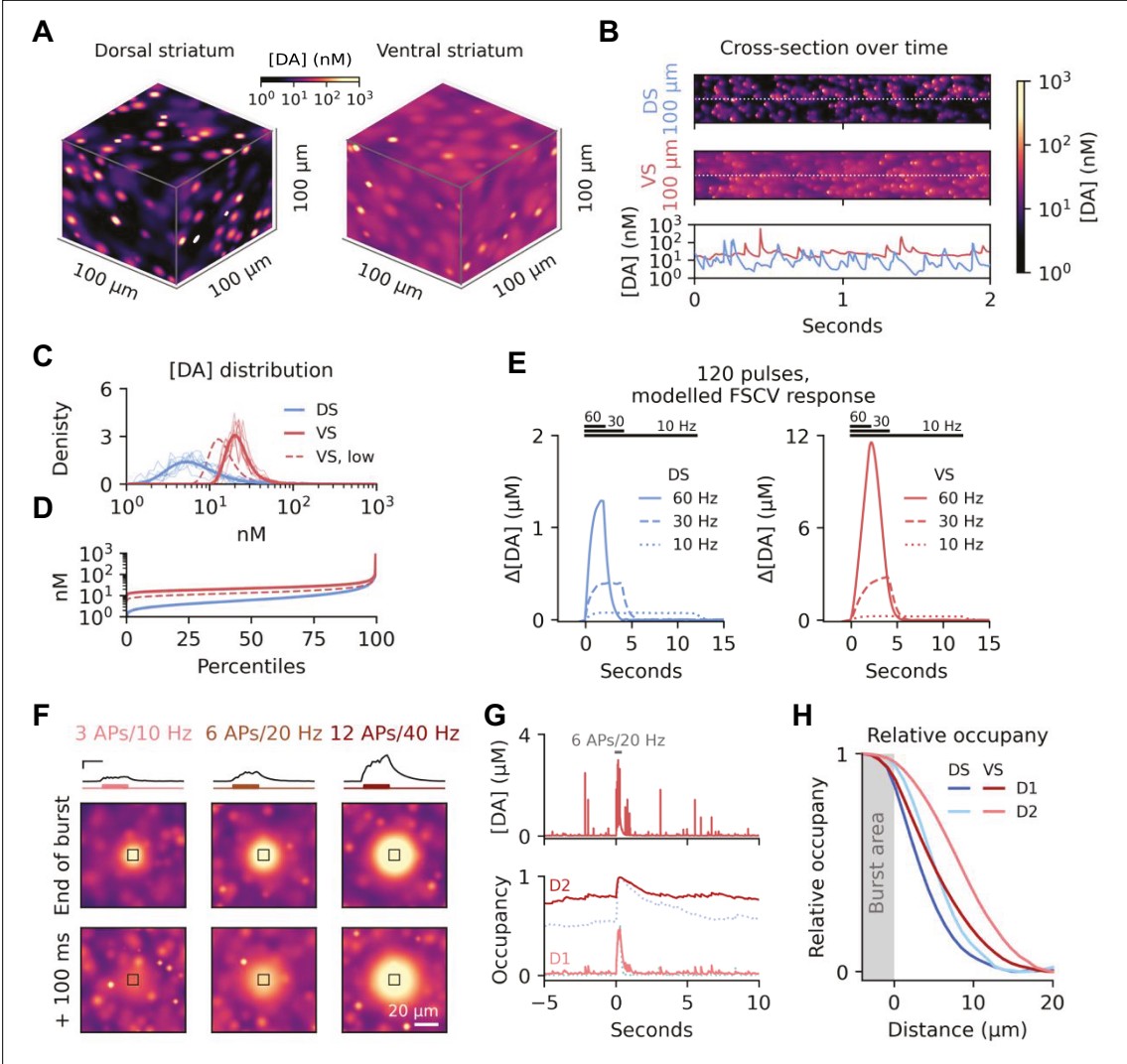

**Figure 2.** Regional differences in uptake greatly impact DA dynamics. (**A**) Representative snapshots of steady state dynamics at 4 Hz tonic firing with parameters mirroring the dorsal (left) and ventral striatum (right). (**B**) Cross-section of temporal dynamics for data shown in a. The bottom row shows concentrations of the dashed lines in the top panels. (**C**) Normalised density of DA concentration of simulations in (**a**). Thick lines are for the entire space, and thin lines are across time for five randomly sampled locations. Dashed red line is for simulation of the ventral striatum with lowest reported innervation density in the literature. (**D**) Same data as in (**c**), but for concentration percentiles. Note that even the lowest percentiles of VS were above 10 nM in [DA]. (**E**) Convolved model response (Figure S1c) to mimic FSCV measurements mirroring the experimentally tested stimulation paradigm in *May and Wightman, 1989* for the dorsal (left) and ventral striatum (right) (**F**) DA release during three burst activity scenarios for all release sites in a 10 x 10 × 10 µm cube (black boxes) and spill-over into the surrounding space. Burst simulated as an increase in firing rate on top of continued tonic firing of the surrounding space. Traces on top are average DA concentrations for the marked cubes, with bursts schematised by coloured lines below. The first image row is at the end of the burst, and the second row is another 100 ms after. Scale bars for traces are 200 ms and 500 nM. Scale bar for the images is 20 µm. (**G**) Top: representative [DA] trace 1 µm away from a release site during pacemaker and burst activity. Bottom: Occupancy of D1Rs and D2Rs for the same site. Occupancy data from the corresponding DS simulation on *Figure 1k* shown as a dotted line. (**H**) Peak occupancy at different distances from the area bursting, normalised to maximal and minimum occupancy.

The online version of this article includes the following source code and figure supplement(s) for figure 2:

**Source code 1.** Source code used to generate data in A-F.

**Source code 2.** Source code used to generate data in G and H.

**Figure supplement 1.** Histochemical gradient of DAT and VMAT2 fluorescence.

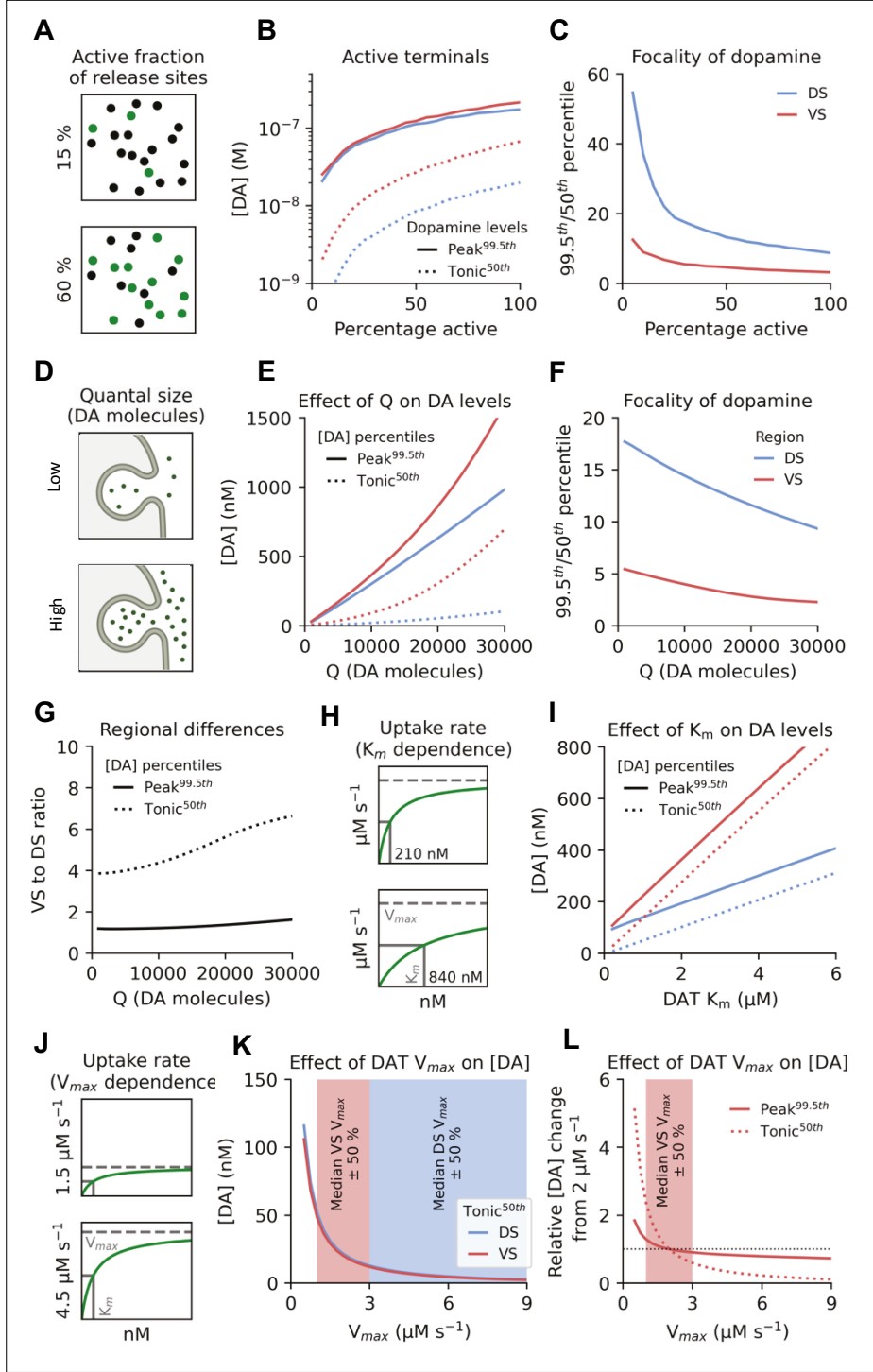

**Figure 3.** Sensitivity of the model to parameter changes. (**A**) Schematic of the fraction of active release sites. Black dots are inactive sites, and green dots indicate actively releasing sites. (**B**) Effect of changing fraction of active release sites on DA concentrations. Blue line, DS peak DA concentration (99.5th percentile); Red line, VS peak DA concentration (99.5th percentile); Dotted blue line, DS tonic DA concentration (50th percentile); Dotted red line, VS tonic DA concentration (50th percentile). (**C**) Ratio between peak (99.5th percentile) and tonic (50th percentile) concentrations across fractions of active release sites in the DS (blue line) and VS (red line) as a measure of DA signal focality. (**D**) Schematic of changing quantal size (Q). (**E**) Effect of changing quantal size on tonic and peak DA concentrations in DS (blue lines) and VS (red lines). (**F**) Ratio between peak and tonic concentrations across various quantal sizes in in DS (blue line) and VS (red line). (**G**) Relative difference between the DS and VS for peak (black

*Figure 3 continued on next page*

*Figure 3 continued*

line) and tonic DA (dotted line) at different quantal sizes. (**H**) Schematic of changing DAT $K_m$. (**i**) Effect of changing DAT $K_m$ on DA concentrations in DS (blue lines) and VS (red lines). (**J**) Schematic of changing DAT $V_{max}$. (**K**) Effect of changing DAT $V_{max}$ on DA concentrations. Shaded areas are median $V_{max}$ of the two regions (DS and VS) as found in the literature shown in ***Appendix 2—table 2*** ± 50%. (**L**) Effect of changing DAT $V_{max}$, with tonic (50[th] percentile) and peak (99.5[th] percentile) DA concentrations normalised to their value at 2 μm s⁻¹ (median value for VS). The shaded area indicates median $V_{max}$ for VS found in the literature shown in ***Appendix 2—table 2*** ± 50%.

The online version of this article includes the following source code and figure supplement(s) for figure 3:

**Source code 1.** Source code used to generate data in B, C, E-G.

**Source code 2.** Source code used to generate data in I.

**Source code 3.** Source code used to generate data in K and L.

**Figure supplement 1.** Model parameter testing.

**Figure supplement 1—source code 1.** Source code used to generate data in ***Figure 3—figure supplement 1***.

**Figure supplement 2.** Fold change during inhibition, $V_{max}$-sensitivity at different release parameters and release-uptake balance.

**Figure supplement 2—source code 1.** Source code used to generate data in ***Figure 3—figure supplement 2A-D***.

---

the DS at a full 100% active sites, underscoring a marked difference in the spatial confinement of DA signals in VS and DS.

The predicted total DA content of a vesicle and the fraction of content released per fusion event is reported to range from 1,000–30,000 molecules (***Garris et al., 1994***; ***Pothos et al., 1998***; ***Staal et al., 2004***; ***Sulzer and Pothos, 2000***; ***Figure 3D***). As expected, tonic and peak concentrations increased in both DS and VS as quantal size was increased (***Figure 3E***). Also, as expected, the focality of the DA distributions dropped for both regions as quantal size increased (***Figure 3F***). The relative difference in tonic DA, however, remained persistently higher in VS and even increased as quantal size increased, indicating a tendency for VS to maintain basal levels of DA regardless of release content (***Figure 3G***). The higher end of quantal sizes, however, resulted in median concentrations far beyond what is typically reported (***Figure 3E***; ***Sulzer et al., 2016***). We observed a largely similar pattern when changing either release probability or firing rate (***Figure 3—figure supplement 1B-G***).

DAT activity is governed by two parameters: $K_m$ and $V_{max}$ (***Kristensen et al., 2011***). To mimic competitive inhibition of DAT by, for example cocaine, we ran a simulation across various $K_m$ values (***Figure 3H***) showing that increasing $K_m$ caused a linear increase in DA levels, consistent with DAT uptake rate responding almost linearly to increases in [DA] below $K_m$ (***Figure 3I***). Of note, most microdialysis studies have reported that cocaine increases [DA] to the same degree in both DS and VSBEsrt wishes; however, these quantifications are usually derived as a ratio of the absolute baseline level (***Carboni et al., 2001***; ***Maisonneuve and Glick, 1992***). If we divide our simulations of increasing $K_m$ with the basal levels estimated in ***Figure 3A-C*** a similar response for DS and VS is found (***Figure 3—figure supplement 2A***). Further, we observe a convergence on a twofold difference in the absolute values at both tonic and peak levels, which matches reports from earlier FSCV studies (***Figure 3—figure supplement 2B***; ***Wu et al., 2001***). This regionally differential response to cocaine matches our observations in a previous biosensor-based study (***Jørgensen et al., 2023***).

Finally, we changed $V_{max}$ by ±50% in both regions and observed a smaller change in tonic level in DS (11 nM) than in VS (38 nM) (***Figure 3K***). This suggests modulation of $V_{max}$ has higher impact in VS than DS. Further, the impact of changing $V_{max}$ in VS was independent of both Q and $R_%$ within values typically reported in the literature (***Figure 3—figure supplement 2C, D***). In contrast to the changes in tonic levels, the relative effect $V_{max}$ had on peak levels was much more modest (***Figure 3L***).

Changes to uptake rate may be mediated by DAT internalisation pathways, but to our knowledge, there is not in vivo evidence of differential release-uptake balances between animals that could lead to varying tonic DA levels across animals. We therefore reanalysed data from our previously published comparison of fibre photometry and microdialysis (***Ejdrup et al., 2023***) and found evidence of natural variations in the release-uptake balance of the mice (***Figure 3—figure supplement 2E, F***), which may underlie different tonic levels of DA in the striatum between animals.

## DAT nanoclustering affects steady state [DA] and clearance after bursts

Our simulations highlight DAT $V_{max}$ as an effective regulator of extracellular DA levels in VS (*Figure 3K*). Internalisation of DAT can serve as a mechanism for this control but is a relatively slow process operating on the order of minutes (*Kristensen et al., 2011*). Interestingly, our recent studies have provided evidence that DAT move laterally in the plasma membrane and transition from a clustered to an unclustered nanoscale distribution in response to excitatory drive and other inputs (*Lycas et al., 2022*; *Rahbek-Clemmensen et al., 2017*). This led us to hypothesise that DAT nanoclustering serves as a mechanism for regulating DAT activity on a faster time scale. We speculated that dense nanoclusters of DAT would produce domains of low [DA] due to uptake overpowering diffusion (*Figure 4A*). As the uptake rate is concentration dependent, this would reduce uptake efficiency (*Figure 4B*). To address this hypothesis, we simulated a single ellipsoid varicosity of 1.5 µm in length and 800 nm thick with surrounding extracellular space (*Ducrot et al., 2021*). The surface was unfolded to a square of equal area (*Figure 4D*), and as 9–16.4% of terminals in the striatum are estimated to be dopaminergic (*Hökfelt, 1968*; *Tennyson et al., 1974*), we set the volume of the surrounding space to seven times the varicosity volume. On the surface of the varicosity, we randomly distributed eight DAT nanoclusters (*Figure 4C*) and ran simulations of how DAT clustering density influenced the DA clearance from the surrounding space. The observed DA concentration in the space surrounding the varicosity shown in *Figure 4C* is illustrated by the cross-section shown in *Figure 4D*. Mean DA uptake capacity of the entire space was kept constant at 4 µM s$^{-1}$ (between the values observed for DS and VS) throughout the simulations, representing a constant amount of DAT molecules on the surface of the varicosity. We only changed the fraction of the surface of the varicosity that was uptake competent by altering the cluster size from small clusters of high density to large clusters of lower density. We ran simulations of eight identical clusters at either 20, 40, 80, or 160 nm in diameter to mirror experimentally observed cluster sizes on DA varicosities, as well as a scenario with DAT fully dispersed (*Figure 4E*; *Lycas et al., 2022*; *Rahbek-Clemmensen et al., 2017*). We first performed a test to see the effect clustering would have at steady state concentrations during pacemaker activity. In the unclustered scenario, average [DA] in the simulation hovered at ~15 nM (*Figure 4F*). However, upon changing to a clustered architecture, [DA] rapidly increased up to 100% (~30 nM) in just 400 ms depending on the degree of clustering (*Figure 4F*). We further wanted to test what effect nanoclustering would have on clearance after burst activity. To do this, we set the extracellular space to a [DA] of 100 nM and performed simulations for the same clustering scenarios (*Figure 4G*). Cluster size dramatically affected clearance time, with the most dense clusters taking almost 400ms to reduce [DA] to 5 nM, compared to just ~200 ms for the unclustered scenario. The hyper-local low-[DA] environment that arose around the dense DAT clusters became apparent when we plotted the [DA] at the centre of a cluster compared to the mean [DA] of the extracellular space (*Figure 4H* – unclustered also showed a drop from surface of varicosity to mean of extracellular space; see general gradient of [DA] in *Figure 4D*). For the 20 nm cluster scenario, the clearance was almost entirely limited by how quickly DA diffused to the nanocluster, as the local [DA] dropped to near zero. We originally hypothesised the effect would mainly be due to a depression in [DA] at the very cluster centre, but a concentration profile of the 80 nm cluster scenario revealed the entirety of the cluster was enveloped in a low-[DA] environment (*Figure 4I*). Accordingly, if DA receptors are located directly next to a dense nanocluster, this effect could feasibly alter the concentration the individual receptors are exposed to.

Summarised, the data supports that DAT nanoclusters produce domains of low [DA] as uptake outcompetes diffusion. As transporter uptake is concentration dependent, the overall uptake efficiency is reduced, which in turn may lead to higher extracellular concentrations of DA. The simulations therefore posit DAT nanoclustering as an efficient way to regulate the extracellular levels of both tonic DA and the spatiotemporal [DA] profiles following release.

## DAT clusters more in the ventral striatum

As our simulations suggest that nanodomain clustering is a way of regulating uptake, and this may be particularly efficient at controlling extracellular DA in VS, we hypothesised nanoscale clustering would be more prominent in VS. To investigate this, we reanalysed data from a previous publication (*Sørensen et al., 2021*), where we acquired super-resolution images of coronally sliced striatal sections from mice stained for DAT. For the present study, images were split into DS and VS based on where in the slices they were taken (*Figure 4J*). DA terminals were identified by vesicular monoamine

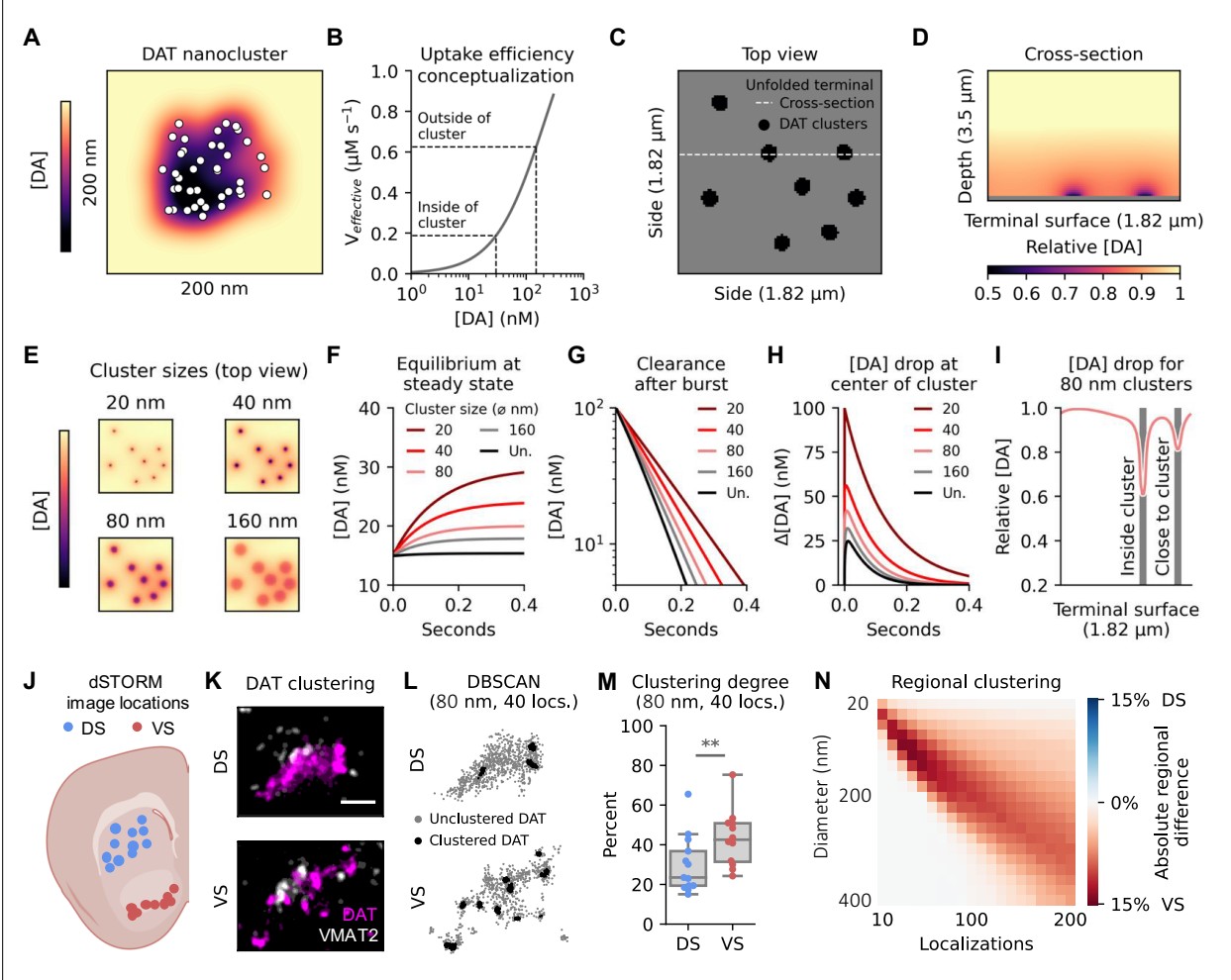

**Figure 4.** DAT nanoclustering reduces uptake and shows regional variation. (**A**) Schematic of dense DA cluster. White dots represent individual DAT molecules, and colour gradient the surrounding DA concentration. (**B**) Effective transport rate dependent on local concentration. (**C**) Top view of unfolded DA varicosity. Black shapes denote clusters of DAT. A dashed white line indicates placement of cross-section shown in (**d**). (**D**) Cross-section showing DA concentration in space surrounding varicosity unfolded in c. The grey line at the bottom is the surface of the varicosity. Colour-coded for DA concentration. (**E**) Top view from (**c**), but colour coded for DA concentration immediately above membrane surface at different DAT cluster sizes. (**F**) Changes in [DA] from 15 nM unclustered (Un.) steady state with constant release after changing to four different cluster size scenarios (ø=diameter). (**G**) Clearance of 100 nM [DA] for different DAT cluster sizes. (**H**) Difference between DA concentration at the centre of clusters (or general surface of varicosity for unclustered) and mean concentration of the full simulation space (**I**) Concentrations across a cross section of a surface with 80 nm diameter clusters. Shaded areas highlight cluster locations. (**J**) Location of images of the dorsal (DS) and ventral striatum (VS) in striatal slices from mice as imaged in *Sørensen et al., 2021* with direct stochastic optical reconstruction microscopy (dSTORM). (**K**) Two representative DA varicosities from DS and VS with VMAT2 in white and DAT in magenta. Images are 1.5x2 µm (scale bar 0.5 µm). (**L**) Individual DAT localisations (locs.) from images in (**k**) coloured by clustering. Black indicates localisation identified as clustered based on DBSCAN with parameters 80 nm diameter and 40 localisations. Grey indicates unclustered localisations. (**M**) Quantification of clustering across all images in (**j**) with parameters in (**l**). Welch's two-sample t-test, p=0.012(*), n=12 (DS) and 13 (VS). (**N**) Absolute difference in percentage of clustering as assessed with DBSCAN across a range of parameters. VS has a higher propensity to cluster across cluster sizes typically reported for DAT clusters.

The online version of this article includes the following source code for figure 4:

**Source code 1.** Source code for simulation.

**Source code 2.** Source code used to generate data in A-E and G-I.

**Source code 3.** Source code used to generate data in F.

transporter 2 (VMAT2) expression, and qualitatively, DAT appeared more clustered in VS than DS (*Figure 4K*). To quantify this, we applied the clustering algorithm Density-Based Spatial Clustering of Applications with Noise (DBSCAN) (*Figure 4L*). This confirmed a regional difference with more DAT localised clusters when using DBSCAN to identify clusters with scanning diameter of 80 nm in VS

compared to DS (*Figure 4L and M*), and we observed a similar regional difference across the range of cluster sizes typically reported (20–200 nm; *Figure 4N*; *Lycas et al., 2022*; *Rahbek-Clemmensen et al., 2017*) supporting the conclusion that DAT nanoclustering is more prevalent in VS than DS.

## Discussion

We developed a three-dimensional, finite-difference computational model to investigate spatio-temporal DA dynamics of striatal subregions in detail. Leveraging prior experimental information on regional differences in dopaminergic innervation density and DAT uptake capacity, our model predicts important differences in dopaminergic dynamics between DS and VS. Strikingly, our simulations suggest that large areas of the DS are effectively devoid of a basal level of DA at pacemaker activity, whereas VS maintain a more homogenous tonic-like basal DA concentration with only small changes in uptake activity powerfully regulating the extracellular DA tone. Furthermore, we modelled receptor binding kinetics and found that D1R binding faithfully followed recently described rapid DA dynamics of the striatum (*Ejdrup et al., 2023*; *Jørgensen et al., 2023*; *Markowitz et al., 2023*), while D2R, with an off-rate of ~5 s, appeared better suited for detecting background tone and integrating prolonged activity (*Howe et al., 2013*; *Jørgensen et al., 2023*). Collectively, these observations have important implications for our understanding of striatal function in behaviour, including decoding of inputs from the prefrontal cortex (PFC) and limbic system as well as the influential phasic-tonic model of dopaminergic signalling (*Grace et al., 2007*; *Niv et al., 2007*; *Schultz, 2007*).

It has been assumed for long that there are tonic levels of DA in the striatum (*Niv et al., 2007*; *Schultz, 2007*; *Sulzer et al., 2016*), although the phenomenon has no clear definition (*Berke, 2018*). Our simulation of DA dynamics in DS during pacemaker activity showed no evidence for a homogenous extracellular distribution. Rather, elevated [DA] was transiently present around release sites during pacemaker activity, with the remaining space mostly depleted of DA. The absence of a general tonic DA level in DS predicted by our model directly supports the notion that DA release sites in DS establish distinct and only partially overlapping DA domains rather than diffuse, tonic DA levels (see *Liu et al., 2021*). This conclusion aligns with recent data where we found that DA concentrations measured by microdialysis correlate with the average of rapid activity recorded with fibre photometry rather than a baseline DA tone (*Ejdrup et al., 2023*). Earlier modelling work by Wickens and colleagues predicted pacemaker activity would generate a tonic, uniform concentration (*Arbuthnott and Wickens, 2007*). But our modelling suggested this is prevented by the significant DA uptake capacity of the DS, as measured by more recent reuptake studies (see *Appendix 2—table 2*).

In contrast to the DS, our model predicted VS to hold a considerable basal level of DA even in spaces without an immediately adjacent release site. This is conceivably what most refer to as tonic DA. In this study, we quantified tonic DA as the median concentration of the entire space (50th percentile), which appeared significantly higher in VS than DS because of the lower VS uptake capacity. Importantly, this matches the results of our direct in vivo comparison of DS and VS in freely moving mice (*Jørgensen et al., 2023*), as well as supporting a spatial gradient of time horizons in the striatum previously predicted in a separate work by *Wickens et al., 2007* and measured in vivo by *Mohebi et al., 2024*.

To challenge our model predictions, we performed our simulations across a wide range of parameters. Only changes to $V_{max}$ for uptake generated differential responses in the two regions. With release and uptake parameters at values from the literature, VS was at a critical point where minor changes to uptake significantly impacted the tonic levels without any major effect on peak concentrations.

Contrary to our observations, some previous microdialysis experiments have suggested higher basal DA levels in the DS compared to VS (*Kuczenski and Segal, 1992*; *Shen et al., 2004*) and have reported two to four times higher [DA] in DS compared to VS (*Kuczenski and Segal, 1992*; *Shen et al., 2004*). However, there are disparate observations in the literature (e.g. *Carboni et al., 2001* reported 20% higher [DA] in VS *Carboni et al., 2001*). Moreover, it is important to note that regional comparisons in microdialysis might be confounded by the considerably higher uptake rate in DS. This will increase the extraction fraction and possibly lead to a significant overestimation of the extracellular concentration as compared to a region with lower uptake rate, such as the VS (*Chefer et al., 2009*).

Our three-dimensional simulations highlight DAT-mediated reuptake as a key mechanism governing striatal DA dynamics and as a key mediator of regional-specific DA dynamics. A physiologically

relevant way to regulate uptake capacity is moving DAT to and from the plasma membrane. Indeed, DAT is subject to such regulation and some of these mechanisms may even be exclusive to the ventral region, including protein kinase C-induced DAT internalisation and Vav2 regulation of DAT surface expression (*Fagan et al., 2020*; *Zhu et al., 2015*). Chemogenetic $G_q$-coupled DREADD activation of DA neurons also results in differential DAT trafficking in the two regions (*Fagan et al., 2020*; *Kearney et al., 2023*). The findings position DAT regulation as an excellent candidate for changing tonic DA levels in VS, which has been proposed to selectively attenuate afferent drive from the PFC through D2R activation (*Grace et al., 2007*). Recent studies of D2R-expressing spiny projection neurons (SPNs) in the VS also suggest that the receptor is not fully saturated under basal firing (*Lee et al., 2020*), matching both our simulations of receptor binding and the notion that tonic DA can be manipulated to alter D2R activation.

If changing uptake capacity is to have a behavioural relevance on a fast timescale, a mechanism to regulate DAT function faster than internalisation must exist. Importantly, the transporter does not only move to and from the surface, but also laterally in the plasma membrane. We have reported that DAT forms nanoclusters in the plasma membrane that dynamically reshape based on excitatory and inhibitory input (*Lycas et al., 2022*; *Rahbek-Clemmensen et al., 2017*). Moreover, we have previously shown that cocaine, which both competitively inhibits DAT and reorganises the transporter nano-domains (*Lycas et al., 2022*), changes the DA signal of the DS to dynamics akin to the VS (*Jørgensen et al., 2023*). Importantly, our simulations showed that nanoclustering may be an effective way to sequester DAT in a dense domain where uptake overpowers diffusion and, as a result, brings down effective uptake speed through local DA depletion. This is in line with evidence that these DAT nano-clusters are enriched in phosphatidylinositol-4,5-bisphosphate (PIP2), and that metabolism of PIP2 decreases uptake rate of DAT (*Lycas et al., 2022*; *Carvelli et al., 2002*). We also found that the nano-clustering phenomenon was considerably more prevalent in VS than in DS. Taken together, these data point to DAT nanoclustering as a way to shape both the spatiotemporal profile of DA release as well as the tonic levels of DA in the striatum – particularly in the VS.

Our incorporation of receptor binding was inspired by important previous modelling work (*Dreyer et al., 2010*; *Dreyer and Hounsgaard, 2013*; *Hunger et al., 2020*). However, earlier models by Dreyer & colleagues assumed instantaneous equilibrium between extracellular DA and receptor occu-pation, which disregards differences in kinetics of the DA receptors that greatly impact transmission dynamics. While later work by Hunger and colleagues introduced more complex receptor modelling, they based their kinetics parameters on early pharmacological studies, whose values likely would prevent DA receptors from decoding signal below the order of minutes (*Burt et al., 1976*; *Maeno, 1982*; *Nishikori et al., 1980*; *Sano et al., 1979*). Instead, we based our receptor kinetics on newer pharmacological experiments in live cells (*Ågren et al., 2021*) and properties of the recently developed DA receptor-based biosensors (*Labouesse and Patriarchi, 2021*), whose receptor values match well despite different methodological approaches. The biosensors are mutated receptors whose kinetics may not be identical to the endogenous receptors, but only the intracellular domains are altered, with no apparent changes of the binding site (*Labouesse and Patriarchi, 2021*). Indeed, these biosensors exhibit kinetics that are well aligned with both modelled and experimentally reported extracellular DA dynamics using non-biosensor-based methods (*Atcherley et al., 2015*; *Gonon et al., 2000*; *Venton et al., 2002*). We believe accordingly that our updated parameters are more accurate portrayals of in vivo conditions; however, as shown throughout the study, the affinity values greatly affect the results. Therefore, we find it important that our model will be available to the research community, allowing others to test their own estimates of receptor kinetics and assess their impact on the model's behaviour.

The presented simulations suggested that receptor binding was largely invariant to single release events during pacemaker activity, while bursts of activity rapidly changed occupancy. Both D1R and D2R immediately responded to burst onset; however, while D1R occupancy rapidly declined to zero within approximately 50 ms, the slow D2R kinetics resulted in an occupancy decline over ~5 s, returning to the baseline maintained by tonic firing. This means that D1R is better suited to discrimi-nate inputs in rapid succession and allow for postsynaptic decoding of the fast-paced in vivo dynamics described particularly for DS (*Ejdrup et al., 2023*; *Jørgensen et al., 2023*; *Markowitz et al., 2023*). By contrast, our analysis shows that D2Rs integrate DA signals over several seconds. As D1R occu-pancy is negligible during pacemaker activity and D2R kinetics are too slow to pick up rapid changes

in DA concentration, our simulations moreover suggest pauses in firing of less than 1 s are not an effective way of signalling for the striatal dopaminergic system. Notably, this finding was apparent even when the D2 affinity was increased an order of magnitude. This challenges the effectiveness of proposed negative reward prediction errors, as even a long pause in firing would have a limited effect on D2 receptor occupation and downstream signalling. This may explain why DA drops during reward omissions are not nearly as prominent as positive signals (*Farrell et al., 2022*; *Greenstreet et al., 2025*).

In conclusion, we have developed a three-dimensional model for DA release dynamics and receptor binding that integrates a wealth of experimentally determined parameters and generates responses to electrical and pharmacological input that fits robustly with literature observations. The model offers an important theoretical framework and a predictive tool that can serve as the basis for future experimental endeavours and help guide the interpretation of new as well as older empirical findings on DA signalling dynamics under both physiological conditions and in disease.

# Materials and methods

### Key resources table

| Reagent type (species) or resource | Designation | Source or reference | Identifiers | Additional information |
|---|---|---|---|---|
| Software, algorithm | Python | https://www.python.org | v3.9.7 | |
| Software, algorithm | Spyder IDE | https://www.spyder-ide.org | v5.1.5 | |
| Software, algorithm | Model code | https://github.com/GetherLab/striatal-dopamine-modelling | | Developed for this work |
| Strain, strain background (*Mus musculus*) | C57Bl/6 J | Details provided in *Sørensen et al., 2021* | | Data used are from *Sørensen et al., 2021* |
| Antibody | anti-DAT Nt (rat monoclonal) | Sigma-Aldrich | MAB369 RRID:AB_2190413 | IF (1:200) *Sørensen et al., 2021* |
| Antibody | Anti-VMAT2 (rabbit polyclonal) | Kind gift from Dr Gary W. Miller, Columbia University *Sørensen et al., 2021* | | IF (1:4000) *Sørensen et al., 2021* |

### Three-dimensional finite difference model

DA release sites in the striatum were simulated at a density of one site per 25 $\mu m^3$ in DS and one site per 27.8 $\mu m^3$ in VS (except for *Figure 4B* and). As a single medium spiny neuron in the striatum is estimated to be influenced by axons from 95 to 194 dopaminergic neurons (*Matsuda et al., 2009*), we simulated DA release sites as originating from 150 separate cell bodies. These were set to fire independently from each other at 4 Hz generated by a Poisson distribution. We set the release probability in response to an action potential at the individual terminal levels to 6% (*Dreyer et al., 2010*), and if a release occurred, the DA concentration of the voxel was elevated by the equivalent of 3000 DA molecules as per *Equation 1*. The volume of each voxel was corrected for an extracellular volume fraction of 0.21 (*Rice and Nicholson, 1991*).

We did not incorporate paired-pulse depression in our model as most in vivo FSCV studies have not reported this phenomenon (*Chergui et al., 1994*; *Garris and Wightman, 1994*; *Hamid et al., 2016*; *May and Wightman, 1989*). In addition, we could explain the gradual blunt of the response during prolonged stimulation (*Figure 2E*) as the result of an equilibrium between release and uptake rather than a depression in release. Newer in vivo biosensor-based studies observe the same 'blunting' phenomenon during stimulation trains (*Mohebi et al., 2023*). A low initial release probability that scales well with frequency has also been reported for VMAT2-positive release sites in dopaminergic cultures where the axons are not severed from their cell bodies as in acute striatal slices (*Silm et al., 2019*). Note that we did not explicitly model autoreceptor inhibition at pacemaker activity. However, this value should be implicitly reflected in the in vivo estimates we base our model on.

When simulating bursts or trains of stimulations, action potentials were added as described for each simulation, but release probability was kept the same unless otherwise noted. Uptake was evenly distributed throughout the simulation space with a $V_{max}$ at 6 $\mu m\ s^{-1}$ for DS and 2 $\mu m\ s^{-1}$ for VS (see *Appendix 2—table 2*) and a $K_m$ at 210 nM (*Hovde et al., 2019*) and calculated for each voxel as per *Equation 2*. For each time step, *dt*, diffusion is calculated with a finite difference implementation of

the Laplacian operator described in *Equation 3* using a diffusion coefficient corrected for tortuosity at 321.7 µm² s⁻¹. Periodic boundaries were used to avoid edge issues with a simulation size of 50 x 50 × 50 µm unless otherwise noted. The Python-based implementation of the model can be found as described in the Code and Data Availability section.

## Receptor occupancy

Our modelling of receptor binding was based on the work of *Dreyer and Hounsgaard, 2013*. Occupancy of the receptors for any given voxel was calculated numerically by numerical integration of:

$$\frac{dD_xR_{occ,t}}{dt} = \left[\text{DA}\right]_t k_{on} \left(1 - D_xR_{occ,t}\right) - k_{off} D_xR_{occ,t} \tag{6}$$

Where $[\text{DA}]_t$ was the dopamine concentration at the given voxel, $D_xR_{occ,t}$ the fraction of DA receptors occupied in the same voxel, $k_{on}$ the rate constant and $k_{off}$ the reverse rate constant. The model assumed no buffering effect from DA bound to receptors, as DA receptors are GPCRs typically expressed at very low levels (*Zhang et al., 2024*).

## Immunohistochemistry analysis

Analysed widefield images were obtained from *Sørensen et al., 2021*. The mouse striatum slices were stained for either DAT or VMAT2 see *Sørensen et al., 2021* for slicing, fixation, staining, and imaging protocol (*Sørensen et al., 2021*). Fluorescence intensities were extracted in a single dorsoventral line parallel to the midline through the horizontally centre of the anterior commissure using ImageJ v1.53q for each slice. Data was vertically centred around the anterior commissure. No background subtraction was performed.

## FSCV modelling

FSCV recordings were modelled by convolving a simulated DA response to a pseudo FSCV trace. The impulse function used was as described in *Venton et al., 2002*. In brief, the impulse function was defined as $\text{IF}_{\text{FSCV}} = e^{-(t+1)\left(k_{-1}*t_s + k_{-2}*t_o\right)}$, where $t$ is time, $t_s$ is scan time, $t_o$ is oxidation time, and $k_{-1}$ and $k_{-2}$ desorption constants. $t_s$ was set to 100 ms as reported in *May and Wightman, 1989* and $t_o$ to 4 ms, which is half of the typically reported symmetrical waveform length. The desorption constants ($k_{-1}$=1.2 s⁻¹ and $k_{-2}$=12 s⁻¹) were taken directly from *Venton et al., 2002*.

## Nanocluster uptake modelling

Nanoclustering of DAT was simulated in Python. An ellipsoid-shaped varicosity of 1.5 µm in length and 800 nm in width was modelled as a flat, square surface with a width of 1.8 µm to match the total surface area (*Lycas et al., 2022*; *Rahbek-Clemmensen et al., 2017*). As 9–16.4% of terminals in the striatum are estimated to be dopaminergic (*Hökfelt, 1968*; *Tennyson et al., 1974*), we set the height of the simulation space to 3.52 µm to generate an extracellular volume seven times the size of the varicosity. Axons and cell bodies are likely a non-negligible part of the striatal volume but omitted in this analysis due to a lack of information in the literature (however, decreasing the share of striatal space taken up by DA varicosities works in the direction of our hypothesis). Spatial granularity was set to 20 nm, and we randomly placed eight non-overlapping locations for clusters on the simulated varicosity. Uptake capacity per voxel for each clustering scenario was set so the mean uptake of the entire space was 4 µm s⁻¹, which resulted in clusters with increased DAT density with decreasing cluster size.

For the steady state simulations, we added [DA] at a constant rate of 224 nM s⁻¹ calculated as follows:

$$\left[DA\right]_{add} = Q_{conc}\, f_{\text{rate}}\, R_\%\, AT_{density} \tag{7}$$

Where $\left[DA\right]_{add}$ was the DA concentration added per second, $Q_{conc}$ the DA concentration added to a voxel by a single release event, $f_{\text{rate}}$ the firing rate at pacemaker activity, $R_\%$ the probability of release for each action potential and $AT_{density}$ the density of actively releasing terminals. $Q_{conc}$ was calculated as follows:

$$Q_{conc} = \frac{Vesicle_{vol}}{Voxel_{vol}} Vesicle_{conc} \frac{1}{EVF} \tag{8}$$

where $Vesicle_{vol}$ is the volume of a vesicle (with an assumed radius of 25 nm), $Voxel_{vol}$ the volume of a voxel in the simulation space, $Vesicle_{conc}$ the concentration of DA in a single vesicle (assuming 3000 DA molecules) and $EVF$ the extracellular volume fraction of the striatum.

For the clearance after burst simulations, we set the initial DA concentration of the entire space to 100 nM.

We ran all the nanoclustering simulations with a time step of $1.875e^{-7}$ s to avoid numerical instability. Diffusion corrected for tortuosity was set to 321.7 µm$^2$ s$^{-1}$ and calculated with a Laplacian operation as described in *Equation 3*. Periodic boundaries were used to account for the varicosity being circular and avoid issues at the edge of the simulation space. The full implementation in Python can be found as described in the Code and Data Availability section.

### Super-resolution clustering analysis

Super-resolved fluorescence microscopy dual-color images from *Sørensen et al., 2021* of striatal brain slices from mice were analysed based on their position in the dorsal or ventral regions (*Figure 4J*). Localisations were fitted using DAOSTORM (*Babcock et al., 2012*) on the originally acquired direct stochastic optical reconstruction microscopy (dSTORM) videos with the following settings: background sigma of 8, maximum likelihood estimation as the fitting error model, 20 peak identification iterations, an initial sigma estimate of 1.5 and a threshold adjusted to each imaging session. Each image was analysed for percentage of localisations within a cluster using density-based spatial clustering of applications with noise (DBSCAN; *Ester et al., 1996*) using the Python package Scikit-learn 0.22 (*Pedregosa et al., 2021*). To uncover broader trends, we scanned across a range of parameters: radius was run from 10 nm to 200 nm and number of localisations from 10 to 200. For *Figure 4M*, a diameter of 80 nm was chosen to match the size reported in *Rahbek-Clemmensen et al., 2017*.

### Statistical analysis

The statistical analyses performed can be found in the legends of each figure. Statistical analyses were performed using open-source python packages SciPy v1.5.2 and NumPy v1.18.1. Boxplots show the 25th and 75th percentile range, and whiskers indicate up to 1.5 times the interquartile range. The remaining data points are plotted as individual outliers.

## Acknowledgements

We thank Dr. Kenneth L Madsen, University of Copenhagen for general input on our modelling and Dr. Nicolas Tritsch, New York University Grossman School of Medicine, for a fruitful discussion on the postsynaptic receptor properties and occupation.

## Additional information

### Competing interests

Jakob Kisbye Dreyer: is affiliated with H. Lundbeck A/S. The author has no other competing interests to declare. The other authors declare that no competing interests exist.

### Funding

| Funder | Grant reference number | Author |
|---|---|---|
| Lundbeck Foundation | R266-2017-4331 | Ulrik Gether |
| Lundbeck Foundation | R276-2018-792 | Ulrik Gether |
| Lundbeck Foundation | R359-2020-2301 | Ulrik Gether |
| Lundbeck Foundation | R181-2014-3090 | Freja Herborg |
| Lundbeck Foundation | R303-2018-3540 | Freja Herborg |
| The Novo Nordisk Foundation | NNF24OC0088870 | Freja Herborg |

| Funder | Grant reference number | Author |
|--------|------------------------|--------|

The funders had no role in study design, data collection and interpretation, or the decision to submit the work for publication.

## Author contributions

Aske Ejdrup, Conceptualization, Data curation, Software, Formal analysis, Validation, Visualization, Methodology, Writing – original draft, Writing – review and editing; Jakob Kisbye Dreyer, Software, Methodology, Writing – review and editing; Matthew D Lycas, Søren H Jørgensen, Investigation, Methodology, Writing – review and editing; Trevor W Robbins, Jeffrey Dalley, Conceptualization, Writing – review and editing; Freja Herborg, Ulrik Gether, Conceptualization, Formal analysis, Supervision, Funding acquisition, Writing – original draft, Project administration, Writing – review and editing

## Author ORCIDs

Aske Ejdrup ⓘ https://orcid.org/0000-0002-4235-7310
Trevor W Robbins ⓘ https://orcid.org/0000-0003-0642-5977
Freja Herborg ⓘ https://orcid.org/0000-0002-0159-4598
Ulrik Gether ⓘ https://orcid.org/0000-0002-0020-3807

Reviewer #1 (Public review): https://doi.org/10.7554/eLife.105214.3.sa1
Reviewer #2 (Public review): https://doi.org/10.7554/eLife.105214.3.sa2
Author response https://doi.org/10.7554/eLife.105214.3.sa3

---

# Additional files

## Supplementary files

MDAR checklist

## Data availability

Numerical implementation and code to run the simulations in a Jupyter notebook format are available at: https://github.com/GetherLab/striatal-dopamine-modelling (copy archived at *Ejdrup, 2024*) or in the source code files accompanying the article in .Py format. The data for *Figure 4* can be simulated with the provided source code (*Figure 4—source code 1*) or found at https://doi.org/10.5281/zenodo.18046987 alongside the data for *Figure 2—figure supplement 1*. The localization files for all dSTORM image analyzed in the paper can also be found at https://doi.org/10.5281/zenodo.18046987. These .txt files contain all the necessary data for the analyses conducted and can be used in combination with the drift correction files also uploaded to recreate the images.

The following dataset was generated:

| Author(s) | Year | Dataset title | Dataset URL | Database and Identifier |
|-----------|------|---------------|-------------|-------------------------|
| Aske E | 2025 | Data for the paper "Computational modelling identifies key determinants of subregion-specific dopamine dynamics in the striatum", Ejdrup et. al, 2025 | https://doi.org/10.5281/zenodo.18046987 | Zenodo, 10.5281/zenodo.18046987 |

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

## Appendix 1

### Supplementary text

Burst release and FSCV modelling

To test the validity of our model, we examined how it would respond to a simulated direct electrical stimulation. We chose this experimental paradigm, as direct stimulation of striatal slices with fast-scan cyclic voltammetry (FSCV) recordings of DA is the most common procedure in the literature to measure DA release. However, only around 10% of the DA release in such a procedure stems from direct stimulation of DA terminals, as 90% can be attributed to cholinergic release from interneurons activating presynaptic nicotinic receptors on the pre-synapse of dopaminergic axons (*Liu et al., 2022*; *Liu et al., 2018*; *Zhou et al., 2001*). This leads to a much stronger response than an action potential travelling up from the midbrain would generate, which is estimated to have an in vivo release probability on the order of 6% (*Dreyer et al., 2010*). We therefore simulated an event with 10 x release probability (60%) to scale to the expected output from an electrical stimulation in slices. The numbers for quantal size vary throughout the literature, but for consistency with previous simulations, we set the value to 3000 DA molecules (*Cragg and Rice, 2004*; *Omiatek et al., 2010*; *Pothos et al., 1998*; *Sulzer and Pothos, 2000*). On the uptake side, DAT has a $V_{max}$ in DS of roughly 6 µM s$^{-1}$ and a $K_m$ of 210 nM (*Appendix 2—table 2*; *Calipari et al., 2012*; *Dreyer et al., 2010*; *Hovde et al., 2019*; *May and Wightman, 1989*; *Salinas et al., 2023*; *Siciliano et al., 2014*). In response to such a stimulation, mean DA concentration ([DA]) at an artificial carbon probe immediately rose to 200 nM with full return to baseline in roughly 0.2 s (Extended Data *Figure 1B and C*). This is significantly faster dynamics than what is typically observed using FSCV, but similar to newer observations made in the DS measured with the dLight biosensors (*Atcherley et al., 2015*; *Brimblecombe et al., 2019*; *Mohebi et al., 2024*; *Salinas et al., 2023*; *Stuber et al., 2010*; *Xie et al., 2020*). The discrepancy between our simulations and biosensor-based recordings versus those recording with FSCV measurements is likely a combination of adsorption rates and the holding period between scans in voltammetry (*Burrell et al., 2015*; *Venton et al., 2002*). To account for this, we inverted the deconvolution processes used in the literature to correct for DA kinetics observed with either voltammetry or amperometry (Extended Data *Figure 1C*; *Atcherley et al., 2015*; *Gonon et al., 2000*; *Venton et al., 2002*). After this process, our results were very similar to those typically observed with FSCV (Fig. S1B; *Atcherley et al., 2015*; *Brimblecombe et al., 2019*; *Stuber et al., 2010*; *Xie et al., 2020*).

# Appendix 2

## Tables 1 and 2

**Appendix 2—table 1.** Overview of reports on dopaminergic density and release in the striatum. Only studies that assessed both regions in rodents are included. Studies that stimulate directly in the striatum are omitted due to the large activation of nicotinic receptors on DA terminals (*1*, *2*). A.U.=arbitrary units, DS = dorsal striatum, VS=ventral striatum, TH = tyrosine hydroxylase.

| Measure | Ratio | DS | VS | Units | Species | Source |
|---|---|---|---|---|---|---|
| TH expression density | 100 % | ~90 | ~90 | A.U. | Mouse | *Alberquilla et al., 2020* |
| TH immunoreactivity | 95 % | ~68 | ~64 | A.U. | Mouse | *Kuroda et al., 2010* |
| TH protein content | 150 % | 0.07 | 0.11 | ng TH/μg prot. | Mouse | *Salvatore et al., 2016* |
| DA content | 90 % | ~155 | ~140 | ng DA/mg prot. | Mouse | *Salvatore et al., 2016* |
| TH immunoreactivity | 75 % | 2.8 | 2.1 | A.U. | Rat | *Huang et al., 2019* |
| TH protein content | 66 % | 0.36 | 0.24 | ng TH/μg prot. | Mouse | *Salvatore et al., 2005* |
| FSCV - $[DA]_p$ | 91 % | 57 | 52 | nM | Rat | *May and Wightman, 1989* |
| FSCV - $[DA]_p$ | 76 % | 89.3 | 67.5 | nM | Rat | *Garris and Wightman, 1994* |
| Median | 90 % | - | - | - | - | - |

**Appendix 2—table 2.** Overview of reported $V_{max}$ values for DA uptake in the striatum. Only studies that assessed both regions in rodents are included. DS = dorsal striatum, VS=ventral striatum, FSCV = fast scan cyclic voltammetry.

| Method | VS/DS Ratio | DS (uM/s) | VS (uM/s) | Species | Source |
|---|---|---|---|---|---|
| FSCV | 29 % | 7.0 | 2.0 | Mouse | *Calipari et al., 2012* |
| FSCV | 28 % | 6.0 | 1.7 | Rat | *Calipari et al., 2012* |
| FSCV | 47 % | 3.0 | 1.4 | Rat | *May and Wightman, 1989* |
| FSCV | 31 % | 6.5 | 2.0 | Mouse | *Siciliano et al., 2014* |
| FSCV | 44 % | 5.0 | 2.2 | Rat | *Ferris et al., 2014* |
| Median | 31 % | 6.0 | 2.0 | - | - |

