## [Editor Report · eLife Assessment]

The conclusions of this work are based on **valuable** simulations of a detailed model of striatal dopamine dynamics. Establishing that lower dopamine uptake rate can lead to a "tonic" level of dopamine in the ventral but not dorsal striatum, and that dopamine concentration changes at short delays can be tracked by D1 but not D2 receptor activation, is invaluable and will be of interest to the community, particularly those studying dopamine. The model simulations provide **convincing** evidence for differences between dorsal and ventral striatum dopamine concentrations, while evidence for differential tracking of dopamine changes by D1 vs D2 receptors is **solid**.

---

## [Referee Report · Reviewer #1 (Public review)]

Ejdrup, Gether and colleagues present a sophisticated simulation of dopamine (DA) dynamics based on a substantial volume of striatum with many DA release sites. The key observation is that reduced DA uptake rate in ventral striatum (VS) compared to dorsal striatum (DS) can produce an appreciable "tonic" level of DA in VS and not DS. In both areas they find that a large proportion of D2 receptors are occupied at "baseline"; this proportion increases with simulated DA cell phasic bursts but has little sensitivity to simulated DA cell pauses. They also examine, in a separate model, the effects of clustering dopamine transporters (DAT) into nanoclusters and say this may be a way of regulating tonic DA levels in VS. I found this work of interest and I think it will be useful to the community.

The conclusion that even an unrealistically long (1s) and complete pause in DA firing has little effect on DA receptor occupancy is potentially very important. The ability to respond to DA pauses has been thought to be a key reason why D2 receptors (may) have high affinity. This simulation instead finds evidence that DA pauses may be useless, from the perspective of reward prediction error signals.

---

## [Referee Report · Reviewer #2 (Public review)]

The work presents a model of dopamine release, diffusion and reuptake in a small (100 micrometer^2 maximum) volume of striatum. This extends previous work by this group and others by comparing dopamine dynamics in the dorsal and ventral striatum and by using a model of immediate dopamine-receptor activation inferred from recent dopamine sensor data. From their simulations the authors report three main conclusions: that ventral and dorsal striatum have consistently different distributions of dopamine; that dorsal striatum does not appear to have a clear "tonic" dopamine -- the sustained, relatively uniform concentration of dopamine driven by the constant 4Hz firing of dopamine neurons; and that D1 receptor activation is able to track rapid increases in dopamine concentration changes D2 receptor activation cannot -- and neither receptor-type's activation tracks pauses in pacemaker firing of dopamine neurons.

The simulations of dorsal striatum will be of interest to dopamine aficionados as they throw doubt on the classic model of "tonic" and "phasic" dopamine actions, further show the disconnect between dopamine neuron firing and consequent release, and thus raise issues for the reward-prediction error theory of dopamine.

There is some careful work here checking the dependence of results on the spatial volume and its discretisation. The simulations of dopamine concentration from pacemaker firing of dopamine neurons are checked over a range of values for key parameters. The model is good, the simulations are well done, and the evidence for robust differences between dorsal and ventral striatum dopamine concentration is good.

There are a couple of weaknesses that suggest further work is needed to support the third conclusion of how DA receptors track dopamine concentration changes, before any strong conclusions are drawn about the implications for the reward prediction error theory of dopamine:

effects of changes in affinity (EC50) are tested, and shown to be robust, but not of the receptors' binding (k_on) and unbinding (k_off) rate constants which are more crucial in setting the ability to track changes in concentration.

bursts of dopamine were modelled as release from a cluster of local release sites (40), which is consistent with induced local release by e.g. cholinergic receptor activation, but the rate of release was modelled as the burst firing of dopamine neurons. Burst firing of dopamine neurons would produce a wide range of release site distributions, and are unlikely to be only locally clustered. Conversely, pauses in dopamine release were seemingly simulated as a blanket cessation of activity at all release sites, which implies a model of complete correlation between dopamine neurons. It would be good to have seen both release scenarios for both types of activity, as well as more nuanced models of phasic firing of dopamine neurons.

That said, in releasing their code openly the authors have made it possible for others to extend this work to test the rate constants, the modelling of dopamine neuron bursting, and more.

---

## [Author Response]

The following is the authors’ response to the original reviews.

**Reviewer #1 (Public review):**
“Ejdrup, Gether, and colleagues present a sophisticated simulation of dopamine (DA) dynamics based on a substantial volume of striatum with many DA release sites. The key observation is that a reduced DA uptake rate in the ventral striatum (VS) compared to the dorsal striatum (DS) can produce an appreciable "tonic" level of DA in VS and not DS. In both areas they find that a large proportion of D2 receptors are occupied at "baseline"; this proportion increases with simulated DA cell phasic bursts but has little sensitivity to simulated DA cell pauses. They also examine, in a separate model, the effects of clustering dopamine transporters (DAT) into nanoclusters and say this may be a way of regulating tonic DA levels in VS. I found this work of interest and I think it will be useful to the community. At the same time, there are a number of weaknesses that should be addressed, and the authors need to more carefully explain how their conclusions are distinct from those based on prior models.

We appreciate that the reviewer finds our work interesting and useful to the community. However, we acknowledge it is important to discuss how our conclusions are different from those reached based on previous model. Already in the original version of the manuscript we discussed our findings in relation to earlier models; however, this discussion has now been expanded. In particular, we would argue that our simulations, which included updated parameters, represent more accurate portrayals of in vivo conditions as it is now specifically stated in lines 466-487. Compared to previous models our data highlight the critical importance of different DAT expression across striatal subregions as a key determinant of differential DA dynamics and differential tonic levels in DS compared to VS. We find that these conclusions are already highlighted in the Abstract and Discussion.

(1) The conclusion that even an unrealistically long (1s) and complete pause in DA firing has little effect on DA receptor occupancy is potentially important. The ability to respond to DA pauses has been thought to be a key reason why D2 receptors (may) have high affinity. This simulation instead finds evidence that DA pauses may be useless. This result should be highlighted in the abstract and discussed more.“

This is an interesting point. We have accordingly carried out new simulations across a range of D2R affinities to assess how this will affect the finding that even a long pause in DA firing has little effect on DR2 receptor occupancy. Interestingly, the simulations demonstrate that this finding is indeed robust across an order of magnitude in affinity, although the sensitivity to a one-second pause goes up as the affinity reaches 20 nM. The data are shown in a revised Figure S1H. For description of the results, please see revised text lines 195-197. The topic is now mentioned in the abstract as well as further commented in the Discussion in lines 500-504.

“(2) The claim of "DAT nanoclustering as a way to shape tonic levels of DA" is not very well supported at present. None of the panels in Figure 4 simply show mean steady-state extracellular DA as a function of clustering. Perhaps mean DA is not the relevant measure, but then the authors need to better define what is and why. This issue may be linked to the fact that DAT clustering is modeled separately (Figure 4) to the main model of DA dynamics (Figures 1-3) which per the Methods assumes even distribution of uptake. Presumably, this is because the spatial resolution of the main model is too coarse to incorporate DAT nanoclusters, but it is still a limitation.”

We agree with the reviewer that steady-state extracellular DA as a function of DAT clustering is a useful measure. We have therefore simulated the effects of different nanoclustering scenarios on this measure. We found that the extracellular concentrations went from approximately 15 nM for unclustered DAT to more than 30 nM in the densest clustering scenario. These results are shown in revised Figure 4F and described in the revised text in lines 337-349.

Further, we fully agree that the spatial resolution of the main model is a limitation and, ideally, that the nanoclustering should be combined with the large-scale release simulations. Unfortunately, this would require many orders of magnitude more computational power than currently available.

“As it stands it is convincing (but too obvious) that DAT clustering will increase DA away from clusters, while decreasing it near clusters. I.e. clustering increases heterogeneity, but how this could be relevant to striatal function is not made clear, especially given the different spatial scales of the models.”

Thank you for raising this important point. While it is true that DAT clustering increases heterogeneity in DA distribution at the microscopic level, the diffusion rate is, in most circumstances, too fast to permit concentration differences on a spatial scale relevant for nearby receptors. Accordingly, we propose that the primary effect of DAT nanoclustering is to decrease the overall uptake capacity, which in turn increases overall extracellular DA concentrations. Thus, homogeneous changes in extracellular DA concentrations can arise from regulating heterogenous DAT distribution. An exception to this would be the circumstance where the receptor is located directly next to a dense cluster – i.e. within nanometers. In such cases, local DA availability may be more directly influenced by clustering effects. Please see revised text in lines 354-362 for discussion of this matter.

“(3) I question how reasonable the "12/40" simulated burst firing condition is, since to my knowledge this is well outside the range of firing patterns actually observed for dopamine cells. It would be better to base key results on more realistic values (in particular, fewer action potentials than 12).”

We fully agree that this typically is outside the physiological range. The values are included in addition to more realistic values (3/10 and 6/20) to showcase what extreme situations would look like.

“(4) There is a need to better explain why "focality" is important, and justify the measure used.”

We have expanded on the intention of this measure in the revised manuscript (please see lines 266-268). Thank you for pointing out this lack of clarification.

“(5) Line 191: " D1 receptors (-Rs) were assumed to have a half maximal effective concentration (EC50) of 1000 nM" The assumptions about receptor EC50s are critical to this work and need to be better justified. It would also be good to show what happens if these EC50 numbers are changed by an order of magnitude up or down.”

We agree that these assumptions are critical. Simulations on effective off-rates across a range of EC50 values has now been included in the revised version in Figure 1I and is referred to in lines 188-189.

“(6) Line 459: "we based our receptor kinetics on newer pharmacological experiments in live cells (Agren et al., 2021) and properties of the recently developed DA receptor-based biosensors (Labouesse & Patriarchi, 2021). Indeed, these sensors are mutated receptors but only on the intracellular domains with no changes of the binding site (Labouesse & Patriarchi, 2021)"

This argument is diminished by the observation that different sensors based on the same binding site have different affinities (e.g. in Patriarchi et al. 2018, dLight1.1 has Kd of 330nM while dlight1.3b has Kd of 1600nM).”

We sincerely thank the reviewer for highlighting this important point. We fully recognize the fundamental importance of absolute and relative DA receptor kinetics for modeling DA actions and acknowledge that differences in affinity estimates from sensor-based measurements highlight the inherent uncertainty in selecting receptor kinetics parameters. While we have based our modeling decisions on what we believe to be the most relevant available data, we acknowledge that the choice of receptor kinetics is a topic of ongoing debate. Importantly, we are making our model available to the research community, allowing others to test their own estimates of receptor kinetics and assess their impact on the model’s behavior. In the revised manuscript, we have further elaborated the rationale behind our parameter choices. Please see revised text in lines in lines 177-178 of the Results section and in lines 481-486 of the Discussion.

“(7) Estimates of Vmax for DA uptake are entirely based on prior fast-scan voltammetry studies (Table S2). But FSCV likely produces distorted measures of uptake rate due to the kinetics of DA adsorption and release on the carbon fiber surface.”

We fully agree that this is a limitation of FSCV. However, most of the cited papers attempt to correct for this by way of fitting the output to a multi-parameter model for DA kinetics. If newer literature brings the Vmax values estimated into question, we have made the model publicly available to rerun the simulations with new parameters.

“(8) It is assumed that tortuosity is the same in DS and VS - is this a safe assumption?”

The original paper cited does not specify which region the values are measured in. However, a separate paper estimates the rat cerebellum has a comparable tortuosity index (Nicholson and Phillips, J Physiol. 1981), suggesting it may be a rather uniform value across brain regions. This is now mentioned in lines 98-99 and the reference has been included.

“(9) More discussion is needed about how the conclusions derived from this more elaborate model of DA dynamics are the same, and different, to conclusions drawn from prior relevant models (including those cited, e.g. from Hunger et al. 2020, etc)”.

As part of our revision, we have expanded the current discussion of our finding in the context of previous models in the manuscript in lines 466-487.

**Reviewer #2 (Public review):**
The work presents a model of dopamine release, diffusion, and reuptake in a small (100 micrometers^2 maximum) volume of striatum. This extends previous work by this group and others by comparing dopamine dynamics in the dorsal and ventral striatum and by using a model of immediate dopamine-receptor activation inferred from recent dopamine sensor data. From their simulations, the authors report two main conclusions. The first is that the dorsal striatum does not appear to have a sustained, relatively uniform concentration of dopamine driven by the constant 4Hz firing of dopamine neurons; rather that constant firing appears to create hotspots of dopamine. By contrast, the lower density of release sites and lower rate of reuptake in the ventral striatum creates a sustained concentration of dopamine. The second main conclusion is that D1 receptor (D1R) activation is able to track dopamine concentration changes at short delays but D2 receptor activation cannot.The simulations of the dorsal striatum will be of interest to dopamine aficionados as they throw some doubt on the classic model of "tonic" and "phasic" dopamine actions, further show the disconnect between dopamine neuron firing and consequent release, and thus raise issues for the reward-prediction error theory of dopamine.There is some careful work here checking the dependence of results on the spatial volume and its discretisation. The simulations of dopamine concentration are checked over a range of values for key parameters. The model is good, the simulations are well done, and the evidence for robust differences between dorsal and ventral striatum dopamine concentration is good.However, the main weakness here is that neither of the main conclusions is strongly evidenced as yet. The claim that the dorsal striatum has no "tonic" dopamine concentration is based on the single example simulation of Figure 1 not the extensive simulations over a range of parameters. Some of those later simulations seem to show that the dorsal striatum can have a "tonic" dopamine concentration, though the measurement of this is indirect. It is not clear why the reader should believe the example simulation over those in the robustness checks, for example by identifying which range of parameter values is more realistic.”

We appreciate that the reviewer finds our work interesting and carefully performed.The reviewer is correct that DA dynamics, including the presence and level of tonic DA, are parameter-dependent in both the dorsal striatum (DS) and ventral striatum (VS). Indeed, our simulations across a broad range of biological parameters were intended to help readers understand how such variation would impact the model’s outcomes, particularly since many of the parameters remain contested. Naturally, altering these parameters results in changes to the observed dynamics. However, to derive possible conclusions, we selected a subset of parameters that we believe best reflect the physiological conditions, as elaborated in the manuscript. In response to the reviewer’s comment, we have placed greater emphasis on clarifying which parameter values we believe reflect the physiological conditions the most (see lines 155-157 and 254-255). Additionally, we have underscored that the distinction between tonic and non-tonic states is not a binary outcome but a parameter-dependent continuum (lines 222-225)—one that our model now allows researchers to explore systematically. Finally, we have highlighted how our simulations across parameter space not only capture this continuum but also identify the regimes that produce the most heterogeneous DA signaling, both within and across striatal regions (lines 266-268).

“The claim that D1Rs can track rapid changes in dopamine is not well supported. It is based on a single simulation in Figure 1 (DS) and 2 (VS) by visual inspection of simulated dopamine concentration traces - and even then it is unclear that D1Rs actually track dynamics because they clearly do not track rapid changes in dopamine that are almost as large as those driven by bursts (cf Figure 1i).”

We would like to draw the attention to Figure 1I, where the claim that D1R track rapid changes is supported in more depth (Figure S1 in original manuscript - moved to main figure to highlight this in the revised manuscript). According to this figure, upon coordinated burst firing, the D1R occupancy rapidly increased as diffusion no longer equilibrated the extracellular concentrations on a timescale faster than the receptors – and D1R receptor occupancy closely tracked extracellular DA with a delay on the order of tens of milliseconds. Note that the brief increases in [DA] from uncoordinated stochastic release events from tonic firing in Figure 1H are too brief to drive D1 signaling, as the DA concentration diffuses into the remaining extracellular space on a timescale of 1-5 ms. This is faster than the receptors response rate and does not lead to any downstream signaling according to our simulations. This means D1 kinetics are rapid enough to track coordinated signaling on a ~50 ms timescale and slower, but not fast enough to respond to individual release events from tonic activity.

“The claim also depends on two things that are poorly explained. First, the model of binding here is missing from the text. It seems to be a simple bound-fraction model, simulating a single D1 or D2 receptor. It is unclear whether more complex models would show the same thing.”

We realize that this is not made clear in the methods and, accordingly, we have updated the method section to elaborate on how we model receptor binding. The model simulates occupied fraction of D1R and D2R in every single voxel of the simulation space. Please see lines 546-555.

“Second, crucial to the receptor model here is the inference that D1 receptor unbinding is rapid; but this inference is made based on the kinetics of dopamine sensors and is superficially explained - it is unclear why sensor kinetics should let us extrapolate to receptor kinetics, and unclear how safe is the extrapolation of the linear regression by an order of magnitude to get the D1 unbinding rate.”

We chose to use the sensors because it was possible to estimate precise affinities/off-rates from the fluorescent measurements. Although there might some variation in affinities that could be attributable to the mutations introduced in the sensors, the data clearly separated D1R and D2R with a D1R affinity of ~1000 nM and a D2R affinity of ~7 nM (Labouesse & Patriarchi, 2021) consistent with earlier predictions of receptor affinities. From our assessment of the literature, we found that this was the most reasonable way to estimate affinities and thereby off-rates. Importantly, the model has been made publicly available, so should new measurements arise, the simulations can be rerun with tweaks to the input parameters. To address the concern, we have also expanded a bit on the logic applied in the updated manuscript (please see lines 177-178).

**Reviewing editor Comments :**
The paper could benefit from a critical confrontation not only with existing modeling work as mentioned by the reviewers, but also with existing empirical data on pauses, D2 MSN excitability, and plasticity/learning.”

We thank both the editor and the reviewers for their suggestions on how to improve the manuscript. We have incorporated further modelling on D1R and D2R response to pauses and bursts and expanded our discussion of the results in relation to existing evidence (please see our responses to the reviewers above and the revised text in the manuscript).

**Reviewer #1 (Recommendations for the authors):**
“(1) Many figure panels are too small to read clearly - e.g. "cross-section over time" plots.”

We agree with the reviewer and have increased the size of panels in several of the figures.

(2) Supplementary Videos of the model in action might be useful (and fun to watch).”

Great idea. We have generated videos of both bursts in the 3D projections and the resulting D1R and D2R occupancy in 2D. The videos are included as supplementary material as Videos S1 and S2 and referred to in the text of the revised manuscript.

” (3) Line 305: " Further, the cusp-like behaviour of Vmax in VS was independent of both Q and R%..."It is not clear what the "cusp" refers to here.”

We agree this is a confusing sentence. We have rewritten and eliminated the use of the vague “cusp” terminology in the manuscript.

” (4) Line 311: "We therefore reanalysed data from our previously published comparison of fibre photometry and microdialysis and found evidence of natural variations in the release-uptake balance of the mice (Figure 5F,G)" This figure seems to be missing altogether.”

The manuscript missed “S” in the mentioned sentence to indicate a supplementary figure. We apologies for the confusion and have corrected the text.

(5) Figure 1:1b: need numbers on the color scale.”

We have added numbers in the updated manuscript.

”1c: adding an earlier line (e.g. 2ms) could be helpful?”

We have added a 2 ms line to aid the readers.

”1d: do the colors show DA concentration on the visible surfaces of the cube or some form of projection?”

The colors show concentrations on the surface. We have expanded the text to clarify this.

”1e: is this "cross-section" a randomly-selected line (i.e. 1D) through the cube?”

The cross-section is midway through the cube. We have clarified this in the text.

”1f: "density" misspelled.”

We thank the reviewer for the keen eye. The error has been corrected.

”1g: color bars indicating stimulation time would be improved if they showed the individual stimulation pulses instead.”

The burst is simulated as a Poisson distribution and individual pulses may therefore be misleading.

” Why does the burst simulation include all release sites in a 10x10x10µm cube? Please justify this parameter choice.1h: "1/10" - the "10" is meaningless for a single pulse, right?”

Yes, we agree.

”1i: is this the concentration for a single voxel? Or the average of voxels that are all 1µm from one specific release site?”

Thank you for pointing out the confusing language. The figure is for a voxel containing a release site (with a voxel size of 1 um in diameter).

The legend seems a bit different from the description in the main text ("within 1µm"). As it stands, I also can't tell whether the small DA peaks are related to that particular release site, or to others.

We have updated the text to clear up the confusing language.

” (6) Figure 2:2h: I'm not sure that the "relative occupancy" normalized measure is the most helpful here.”

We believe the figure aids to illustrate the sphere of influence on receptors from a single burst is greater in VS than DS, suggesting DS can process information with tighter spatial control. Using a relative measure allows for more accessible comparison of the sphere of influence in a single figure.

” (7) Figure 3:The schematics need improvement.3a – would be more useful if it corresponded better to the actual simulation (e.g. we had a spatial scale shown).3d – is this really useful, given the number of molecules shown is so much lower than in the simulation?3h, 3j – need more explanation, e.g. axis labels. ”

The schematics are intended to quickly inform the readers what parameters are tuned in the following figures, and not to be exact representations. However, we agree Figures 3h and 3j need axis labels, and we have accordingly added these.

(8) Figure 4:4m, n were not clearly explained.

We agree and have elaborated the explanation of these figures in the manuscript (lines 374-377).

” (9) From Figure S1 it appears that the definition of "DS" and "VS" used is above and below the anterior commissure, respectively. This doesn't seem reasonable - many if not most studies of "VS" have examined the nucleus accumbens core, which extends above the anterior commissure. Instead, it seems like the DAT expression difference observed is primarily a difference between accumbens Shell and the rest of the striatum, rather than DS vs VS.”

We assume that the reviewer refers to Figure S3 and not S1. First, we would like to highlight that we had mislabeled VMAT2 and DAT in Figure S3C (now corrected). Apologies for the confusion. Second, as for striatal subregions, we have intentionally not distinguished between different subregions of the ventral striatum. The majority of literature we base our parameters on do not specify between e.g., NAcC vs. NAcS or DLS vs. DMS. The four slices we examined in Figure 3A-C were not perfectly aligned in the accumbal region, and we therefore do not believe we can draw any conclusions between core and shell.

**Reviewer #2 (Recommendations for the authors):**
(1) Modelling assumptions:The burst activity simulations seem conceptually flawed. How were release sites assigned to the 150 neurons? The burst activity simulations such as Figure 1g show a spatially localised release, but this means either (1) the release sites for one DA neuron are all locally clustered, or (2) only some release sites for each DA neuron are receiving a burst of APs, those release sites are close together, and the DA neurons' other release sites are not receiving the burst. Either way, this is not plausible.”

We apologize for the confusion; however, we disagree that the simulations seem conceptually flawed. It is important to note that the burst simulation is spatially restricted to investigate local DA dynamics and how well different parts of the striatum can gate spill-over and receptor activation. The conditions may mimic local action potentials generated by nicotinic receptor activation (see e.g. Liu et al. Science 2022 or Matityahu et al, Nature Comm 2023), We have accordingly expanded on this is the manuscript on lines 148-151.

(2) Data and its reporting:Comparison to May and Wightman data: if we're meant to compare DS and VS concentrations, then plot them together; what were the experimental results (just says "closely resembled the earlier findings")?”

Unfortunately, the quantitative values of the May and Wightman (1989) data are not publicly available. We are therefore limited to visual comparison and cannot replot the values.

” Figures S3b and c do not agree: Figure S3b shows DAT staining dropping considerably in VS; Fig 3c does not, and neither do the quoted statistics.”

We had accidentally mixed up the labels in Figure S3c. Thank you for spotting this. We have corrected this in the updated manuscript.

” How robust are the results of simulations of the same parameter set? Figures S3D and E imply 5 simulations per burst paradigm, but these are not described.”

The bursts are simulated with a Poisson distribution as described in Methods under Three-dimensional finite difference model. This induces a stochastic variation in the simulations that mimics the empirical observations (see Dreyer et al., J. Neurosci., 2010).

” I found it rather odd that the robustness of the receptor binding results is not checked across the changes in model parameters. This seems necessary because most of the changes, such as increasing the quantal release or the number of sites, will obviously increase dopamine concentration, but they do not necessarily meaningfully increase receptor activation because of saturation (and, in more complex receptor binding models, because of the number of available receptors).”

This is an excellent point. However, we decided not to address this in the present study as we would argue that such additional simulations are not a necessity for our main conclusions. Instead, we decided in the revised version to focus on simulations mirroring a range of different receptor affinities as described in detail above.

” Figure 4H: how can unclustered simulations have a different concentration at the centre of a "cluster" than outside, when the uptake is homogenous? Why is clustering of DAT "efficient"? [line 359]”

This is a great observation. The drop is compared to the average of the simulation space. Despite no clusters, the uniform scenario still has a concentration gradient towards the surface of the varicosity. We have elaborated on this in the manuscript on lines 346-349.

” The Discussion conclusions about what D1Rs and D2Rs cannot track are not tested in the paper (e.g. ramps). Either test them or make clear what is speculation.”

An excellent point that some of the claims in the discussion were not fully supported. We have added a simulation with a chain of burst firings to highlight how the temporal integration differs between the two receptors and updated the wording in the discussion to exclude ramps as this was not explicitly tested. See lines 191-193 and Figure S1G.

” (3) Organisation of paper:Consistency of terminology. These terms seem to be used to describe the same thing, but it is unclear if they are: release sites, active terminals (Table 1), varicosity density. Likewise: release probability, release fraction.”

Thank you for pointing this out. We have revised the manuscript and cleared up terminology on release sites. However, release probability and release-capable fraction of varicosities are two separate concepts.

” The references to the supplementary figure are not in sequence, and the panels assigned to the supplemental figures seem arbitrary in what is assigned to each figure and their ordering. As Figures 1 and 2 are to be directly compared, so plot the same results in each. Figure S1F is discussed as a key result, but is in a supplemental figure. ”

Thank you for identifying this. We have updated figure references and further moved Figure S1F into the main as we agree this is a main finding.

” The paper frequently reads as a loose collection of observations of simulations. For example, why look at the competitive inhibition of DA by cocaine [Fig 3H-I]? The nanoclustering of DAT (Figure 4) seems to be partial work from a different paper - it is unclear why the Vmax results warrant that detailed treatment here, especially as no rationale is offered for why we would want Vmax to change.”

We apologize if the paper reads as a loose collection of observations of simulations. This is certainly not the case. As for the cocaine competition, we used this because this modulates the Km value for DA and because we wanted to examine how dependent the dopamine dynamics are to changing different parameters in the model (Km in this case). We noticed Vmax had a separate effect between DS and VS. Accordingly, we gave it particular focus because it is physiological parameter than be modified and, if modified, it can have potential large impact on striatal DA dynamics. Importantly, it is well known that the DA transporter (DAT) is subject to cellular regulation of its surface expression e.g. by internalization /recycling and thereby of uptake capacity (Vmax). Furthermore, we demonstrate in the present study evidence that uptake capacity on a much faster time scale can be modulated by nanoclustering, which posits a potentially novel type of synaptic plasticity. We find this rather interesting and decided therefore to focus on this in the manuscript.

” What are the axes in Figure 3H and Figure 3J?”

We have updated the figures to include axis. Thank you for pointing out this omission.

” Much is made of the sensitivity to Vmax in VS versus DS, but this was hard work to understand. It took me a while to work out that Figure 3K was meant to indicate the range of Vmax that would be changed in VS and DS respectively. "Cusp-like behaviour" (line 305) is unclear.”

We agree that the original language was unclear – including the terminology “cusplike behavior”. We have updated the description and cut the confusion terminology. See line 366.

” The treatment of highly relevant prior work, especially that of Hunger et al 2020 and Dreyer et al (2010, 2014), is poor, being dismissed in a single paragraph late in the Discussion rather than explicating how the current paper's results fit into the context of that work. The authors may also want to discuss the anticipation of their conclusions by Wickens and colleagues, including dopamine hotspots (https://doi.org/10.1016/j.tins.2006.12.003) and differences between DS and VS dopamine release (https://doi.org/10.1196/annals.1390.016).”

We thank the reviewer for the suggested discussion points and have included and discussed references to the work by Wickens and colleagues (see lines 407-411 and 418-420).

” (4) Methods:Clarify the FSCV simulations: the function I_FSCV was convolved with the simulated [DA] signal?”

Yes. We have clarified this in the method section on lines 593-594.